# Rapid genome editing by CRISPR-Cas9-POLD3 fusion

**Ganna Reint[1†], Zhuokun Li[1†], Kornel Labun[2], Salla Keskitalo[3], Inkeri Soppa[1,4], Katariina Mamia[1], Eero Tolo[5], Monika Szymanska[1], Leonardo A Meza-Zepeda[6], Susanne Lorenz[6], Artur Cieslar-Pobuda[1,7], Xian Hu[1], Diana L Bordin[8], Judith Staerk[1,9], Eivind Valen[2], Bernhard Schmierer[10], Markku Varjosalo[3], Jussi Taipale[10,11,12], Emma Haapaniemi[1,4,13]***

[1]Centre for Molecular Medicine Norway, University of Oslo, Oslo, Norway; [2]Department of Informatics, Computational Biology Unit, University of Bergen, Bergen, Norway; [3]Center for Biotechnology, University of Helsinki, Helsinki, Finland; [4]Stem Cells and Metabolism Research Program, Faculty of Medicine, University of Helsinki, Oslo, Finland; [5]Faculty of Social Sciences, University of Helsinki, Oslo, Finland; [6]Department of Core Facilities, Institute for Cancer Research, Oslo University Hospital, Oslo, Norway; [7]Department of Cancer Immunology, Institute of Cancer Research, Oslo University Hospital, Oslo, Norway; [8]Department of Clinical Molecular Biology, Akershus University Hospital, Oslo, Norway; [9]Department of Haematology, Oslo University Hospital, Oslo, Norway; [10]Department of Medical Biochemistry and Biophysics, Karolinska Institute, Stockholm, Sweden; [11]Department of Biochemistry, University of Cambridge, Cambridge, United Kingdom; [12]Genome-Scale Biology Program, University of Helsinki, Oslo, Norway; [13]Department of Pediatrics, Oslo University Hospital, Oslo, Norway

**\*For correspondence:**
emma.haapaniemi@ncmm.uio.no

†These authors contributed equally to this work

**Competing interest:** The authors declare that no competing interests exist.

**Abstract** Precision CRISPR gene editing relies on the cellular homology-directed DNA repair (HDR) to introduce custom DNA sequences to target sites. The HDR editing efficiency varies between cell types and genomic sites, and the sources of this variation are incompletely understood. Here, we have studied the effect of 450 DNA repair protein-Cas9 fusions on CRISPR genome editing outcomes. We find the majority of fusions to improve precision genome editing only modestly in a locus- and cell-type specific manner. We identify Cas9-POLD3 fusion that enhances editing by speeding up the initiation of DNA repair. We conclude that while DNA repair protein fusions to Cas9 can improve HDR CRISPR editing, most need to be optimized to the cell type and genomic site, highlighting the diversity of factors contributing to locus-specific genome editing outcomes.

## Editor's evaluation

This is a well designed and well conducted study comparing ~450 human DNA repair protein and protein fragments in fusions with CRISPR/Cas9 for their efficiency in HDR genome editing and found that Cas9-POLD3 performed best in screening system or local targeting. The work proved that Cas9-POLD3 enhances editing at the early time points by speeding up the kinetics of Cas9 DNA binding and dissociation, allowing the rapid initiation of DNA repair. This work will be of interest to those wanting to improve gene editing efficiency by HDR.

## Introduction

CRISPR/Cas9 has become a common genome editing tool in both basic and medical research. It induces targeted DNA double-strand breaks (DSBs) that are commonly repaired by non-homologous end-joining (NHEJ), which leads to gene disruption and knockout. The less common pathways utilize homology-directed repair (HDR), which can induce custom genetic changes to the DSB sites by using exogenous DNA as a template. HDR is confined to the synthesis (S) and G2 phases of the cell cycle (*Essers et al., 2002*). Therefore, its efficiency varies between cell types (*Miyaoka et al., 2016*) and improves upon rapid cell proliferation (*Roth et al., 2018*) and upon targeting the Cas9 expression to S cell cycle phase (*Yang et al., 2018*). Factors that impair cell cycle progression from G1 to S decrease CRISPR-Cas9-mediated HDR (*Haapaniemi et al., 2018*). HDR can be promoted by increasing the local concentration of the repair template (*Savic et al., 2018*) and by general NHEJ pathway inhibition (*Maruyama et al., 2015*, *Robert et al., 2015*, *Riesenberg and Maricic, 2018*), which, however, often negatively affects the cell viability and fitness (*Gu et al., 1997*, *O'Driscoll et al., 2001*). In addition, recent studies show that providing the DNA repair template with long homology arms by a non-integrating rAAV6 can significantly increase HDR rates (*Wiebking et al., 2021*, *Lattanzi et al., 2021*). Overall, genome editing is most efficient in open chromatin (*Chen et al., 2017*) and when Cas9 is bound to an actively transcribed DNA strand where the approaching polymerase can rapidly remove the complex (*Clarke et al., 2018*, *Richardson et al., 2016*, *Jones et al., 2017*). Cas9 binds to the DNA for extended periods (*Richardson et al., 2016*, *Jones et al., 2017*) and successful cut/repair requires the activation of pathways typically involved in resolving stalled DNA replication forks (*Richardson et al., 2018*).

Enhancing CRISPR-based HDR by Cas9 fusion proteins can stimulate correction locally at the cut site, without causing generalized disturbance in the cellular DNA repair process and thus increasing the safety and specificity of the editing. HDR-based precision genome editing can be enhanced by fusing Cas9 with DNA repair proteins or their parts (*Tran et al., 2019*, *Charpentier et al., 2018*), chromatin modulating peptides (*Ding et al., 2019*), or peptides that locally decrease NHEJ pathway activation at the DNA double-strand break sites (*Jayavaradhan et al., 2019*). The effect of the published fusions has been guide-, cell type-, and locus-specific, reaching up to 30% (fourfold) HDR for Cas9-HN1HG1 (*Ding et al., 2019*), 20% (sixfold) HDR for Cas9CtIP (*Tran et al., 2019*; *Charpentier et al., 2018*), and 86% (threefold) for Cas9-DN1S (*Jayavaradhan et al., 2019*), depending on the model system used.

Here, we systematically characterize the effect of ~450 human DNA repair protein and protein fragment fusions on CRISPR-Cas9-based HDR genome editing, using reporter human embryonic kidney (HEK293T) and immortalized retinal pigment epithelium (RPE-1) cell models. We note that for most of the proteins, their effect on the HDR editing of the reporter locus is modest. For the subset that improves genome editing, the effect is locus-specific and comparable to previous publications that have studied single Cas9 fusions (*Tran et al., 2019*; *Charpentier et al., 2018*; *Ding et al., 2019*; *Jayavaradhan et al., 2019*). We identify Cas9 fusion to DNA polymerase delta subunit 3 (Cas9-POLD3) as a new HDR enhancer. The fusion enhances editing at the early time points by speeding up the kinetics of Cas9 DNA binding and dissociation, allowing the rapid initiation of DNA repair.

## Results

### The effect of DNA repair protein fusions on CRISPR-Cas9 genome editing

We sought to better understand the processes that govern the choice and efficiency of DNA repair in CRISPR-Cas9-induced breaks. To this end, we conducted arrayed screens where we fused ~450 human proteins and protein domains involved in DNA repair to the C-terminus of wild-type (WT) *Streptococcus pyogenes* Cas9 (Cas9WT, *Figure 1a, b*, *Figure 1—figure supplement 1*, *Supplementary file 1*). The DNA repair protein ORFs were cloned to a Gateway-compatible Cas9 destination plasmid, and the plasmids and a repair DNA template transfected to a clonal HEK293T reporter cell line. The line contains a non-functional green fluorescent protein (GFP), a guide targeting the mutant GFP (sgGFP), and functional blue fluorescent protein (BFP) as a positive selection marker. The proportion of cells that turned GFP positive was used as a proxy for HDR efficiency.

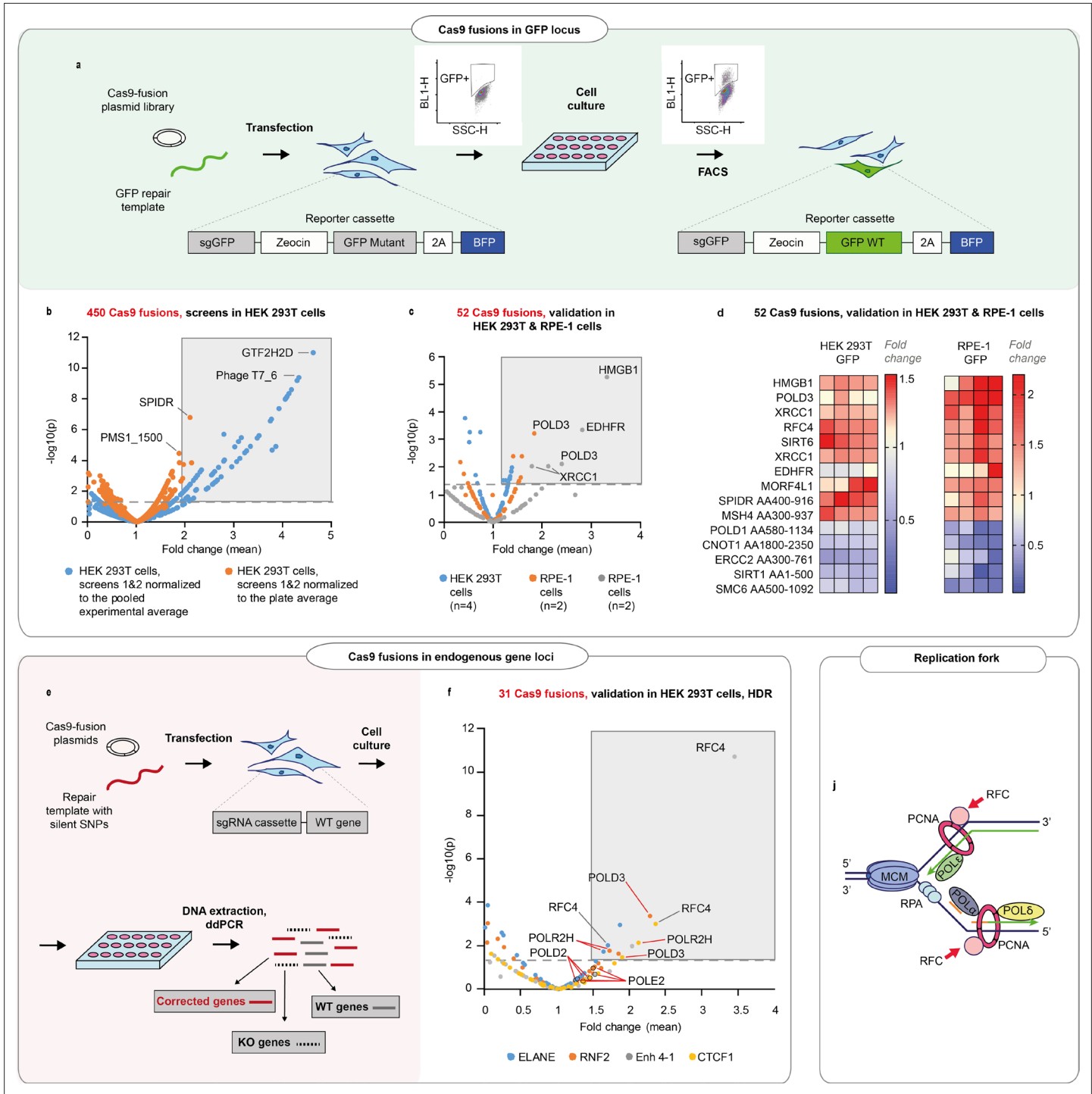

**Figure 1.** DNA repair proteins that affect genome editing outcomes. (**a**) Schematic representation of the screen. Human embryonic kidney (HEK293T) cell line contains a reporter cassette expressing a selection marker, guide (sgGFP) and a mutant GFP-BFP protein. The line is transfected with a GFP repair oligo and an arrayed plasmid library containing ~450 DNA repair proteins that are fused to the C-terminus of Cas9WT. The homology-directed repair (HDR) editing efficiency for each fusion is defined by the percentage of cells with restored GFP function in each transfected well (measured by FACS). (**b**) Normalized GFP recovery values from two independent screens. Both experiments n = 3, each data point represents an average of value of all six replicas (n = 2 × 3), where each replicate was normalized either to the experimental average (blue dots) or to the plate average (orange dots). p-Values were calculated by one-way ANOVA test, where the mean of each triplicate is compared to the combined mean of all other triplicates from the screen. (**c**) Experiment average-normalized GFP recovery values for 52 fusions, chosen based on their performance in panel (**b**). In HEK293T n = 4, one independent biological experiment, in retinal pigment epithelium (RPE-1), n = 2, two independent biological experiments. Statistical significance is

*Figure 1 continued*

calculated as in (**b**). (**d**) Normalized GFP recovery values for 10 best-performing and five worst-performing Cas9 fusions from panel (**c**). Protein fragments are denoted based on the position in canonical transcript (i.e. SPIDR AA400-916 corresponds to a fragment of SPIDR protein which starts at amino acid 400 and ends at amino acid 916). (**e**) Schematic representation of the experiment shown in **f**. (**f**) Experiment average-normalized HDR editing for 31 fusions in four endogenous loci (ELANE, RNF2, Enh 4–1, and CTCF1) in HEK293T cells, quantified by droplet digital PCR. Polymerase fusions are marked with red pointers. n = 3, one independent biological experiment for each locus. Statistical significance is calculated as in (**b**). The raw data points are visible in *Figure 1—figure supplements 3–6* . (**j**) Replication fork schematics. For all experiments, Cas9 was delivered as plasmid, the repair template as single-stranded oligonucleotide, and the guide was expressed from the genome.

The online version of this article includes the following figure supplement(s) for figure 1:

**Figure supplement 1.** Plasmid sizes and molar quantities of plasmids used in the high-throughput screens.

**Figure supplement 2.** Homology-directed repair (HDR) efficiency of the best-performing fusions.

**Figure supplement 3.** HDR efficiency of the best-performing fusions: % of HDR, % HDR of total editing and % of NHEJ in HEK293T cells that stably express the guide targeting CTCF1 endogenous locus.

**Figure supplement 4.** HDR efficiency of the best-performing fusions: % of HDR, % HDR of total editing and % of NHEJ in HEK293T cells that stably express the guide targeting ELANE endogenous locus.

**Figure supplement 5.** HDR efficiency of the best-performing fusions: % of HDR, % HDR of total editing and % of NHEJ in HEK293T cells that stably express the guide targeting Enh 4-1 endogenous locus.

**Figure supplement 6.** HDR efficiency of the best-performing fusions: % of HDR, % HDR of total editing and % of NHEJ in HEK293T cells that stably express the guide targeting RNF2 endogenous locus.

After the initial screen, we chose to validate Cas9WT fusions with >4% GFP+ cells per well or HDR improvement >2 times relative to experiment average (total 52 fusions from two independent screens) (*Figure 1b*, and Materials and methods). We transfected the Cas9 fusion plasmids to clonal reporter HEK293T and RPE-1 (hTERT immortalized retinal pigment epithelium) cell lines. RPE-1 has a near-normal karyotype and is the target cell type in eye gene therapy.

We found that in the GFP reporter locus, the proteins that improved HDR above the experiment mean were largely similar between RPE-1 and HEK293T cells, with Cas9-HMGB1, a previously published editing-enhancing fusion (*Ding et al., 2019*), among the good performers (*Figure 1c, d* and *Figure 1—figure supplement 2a*). We chose to further validate 31 fusions that performed comparably in these two cell lines. We transfected the fusion plasmids to four different HEK293T cell lines that expressed a guide targeting three transcribed and one non-transcribed endogenous loci, respectively. We quantified the HDR and NHEJ editing by droplet digital PCR (ddPCR) (*Figure 1e, f* and *Figure 1—figure supplement 2b*, *Figure 1—figure supplements 3–6*,). ddPCR is a sensitive and quantitative method to detect HDR and NHEJ at endogenous gene loci after gene editing (*Miyaoka et al., 2018*, *Miyaoka et al., 2016*), and the results are comparable to next-generation sequencing-based genome editing quantification (*Figure 2—figure supplement 1k,l*). We discovered several polymerases among the better-performing fusions (POLD3, POLD2, POLR2H, POLE2) (*Figure 1—figure supplement 2b*, *Figure 1—figure supplements 3–6*). In addition, the replication factor C subunits RFC4 and RFC5 performed well in most of the tested loci. POLD3, POLE2, RFC4, and RFC5 are all members of the DNA replication fork machinery *Figure 1i*.

We chose seven Cas9 fusions for the final validation and transfected them as mRNA to human hTERT immortalized fibroblast lines (BJ-5ta) that each expressed a guide targeting five different endogenous loci, respectively (*Figure 2a, b*, *Figure 2—figure supplement 1a-j*). The fusion performance was strongly dependent on individual loci, with POLD3 outperforming the other fusions across the majority of the tested conditions (*Figure 2b*, *Figure 2—figure supplement 1a-j*). We further tested the performance of Cas9-POLD3 across decreasing concentrations in the RPE-1 reporter cells and noted that in all four different concentrations, Cas9-POLD3 outperformed Cas9WT (*Figure 2—figure supplement 2*).

## Cas9-POLD3 fusion accelerates the initiation of cut repair

POLD3 is a component of the replicative polymerase δ, and we hypothesized that POLD3 fusion improves genome editing by speeding up the removal of Cas9 from the cut site (*Clarke et al., 2018*). The faster Cas9 removal results in early recruitment of the DNA damage response machinery to the break and speeds up the DNA repair progression.

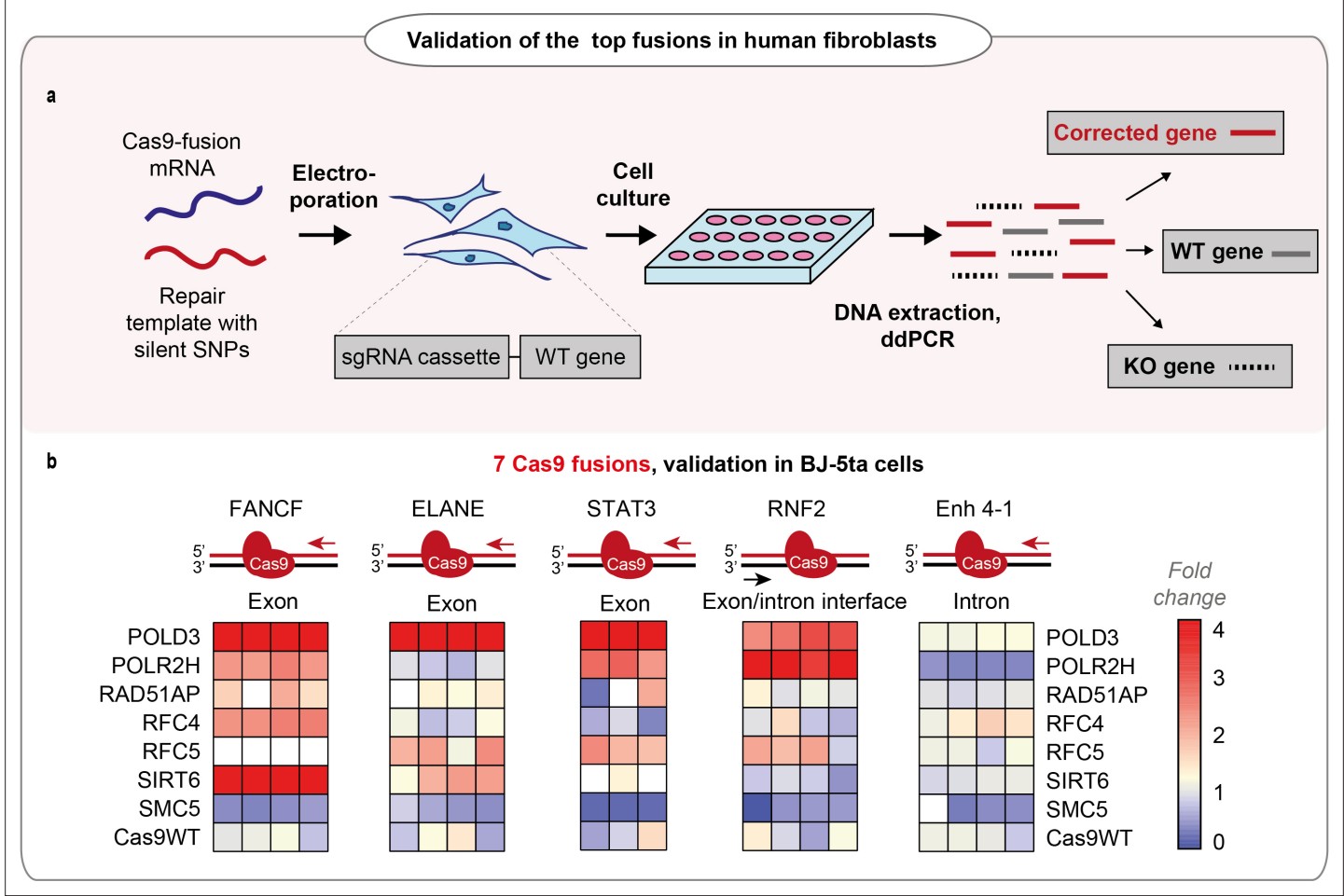

**Figure 2.** Performance of the fusions in BJ-5ta cell line. (**a**) Schematic representation of the experiment shown in b. (**b**) Normalized homology-directed repair (HDR) efficiency of the seven best-performing fusions in five endogenous loci in hTERT immortalized fibroblasts (BJ-5ta), quantified by droplet digital PCR (ddPCR). Heat maps represent values normalized to Cas9WT for each gene locus. The sets have an upper limit threshold and values above it are color-coded as a scale maximum (bright-red). One independent experiment for each gene locus, n = 4 for FANCF, Enh 4–1, RNF2, ELANE gene loci; n = 3 for STAT3 gene. The raw data points are visible in *Figure 2—figure supplement 1a-j*. For all experiments, Cas9 was delivered as mRNA, the repair template as single-stranded oligonucleotide, and the guide was expressed from the genome.

The online version of this article includes the following figure supplement(s) for figure 2:

**Figure supplement 1.** Cas9 fusion editing efficiency in immortalized fibroblasts (BJ-5ta) that stably express sgRNA targeting the indicated loci.

**Figure supplement 2.** Cas9 fusion to DNA polymerase delta subunit 3 (Cas9-POLD3) editing in decreased concentrations.

To understand whether the speed of the overall DSB DNA repair is increased after Cas9-POLD3 editing, we transfected eight fusion plasmids in the RPE-1 reporter cells and quantified the HDR and NHEJ repair across time by ddPCR (*Figure 3a,b*). For Cas9WT and the other fusions, there was a linear increase in successful editing toward later (72 hr) time points. For Cas9-POLD3, the editing peaked earlier (24 hr) and plateaued at the later (72 hr) time point. To further quantify how rapidly the DNA damage response activates upon cutting the DNA, we compared the emergence of the early-stage DNA damage marker γH2AX (*Mah et al., 2010*) which arrives at DNA double-strand breaks within minutes after CRISPR cutting (*Figure 3c-e*). We transfected BJ-5ta fibroblasts with Cas9WT or Cas9-POLD3 mRNA and a pool of four synthesized or expressed guides targeting three coding (RNF2, STAT3, Enh 4–1) and one non-coding locus (CTCF1) and quantified the γH2AX foci with immunofluorescence microscopy. For Cas9WT, the number of foci steadily increased and peaked at 24 hr, after which the count declined. For Cas9-POLD3 the number of foci was already high in the early time points (*Figure 3e*), suggesting rapid Cas9-POLD3 recruitment and removal from the DNA. In parallel to γH2AX foci counting, we quantified the NHEJ progression across time by ddPCR in each individual

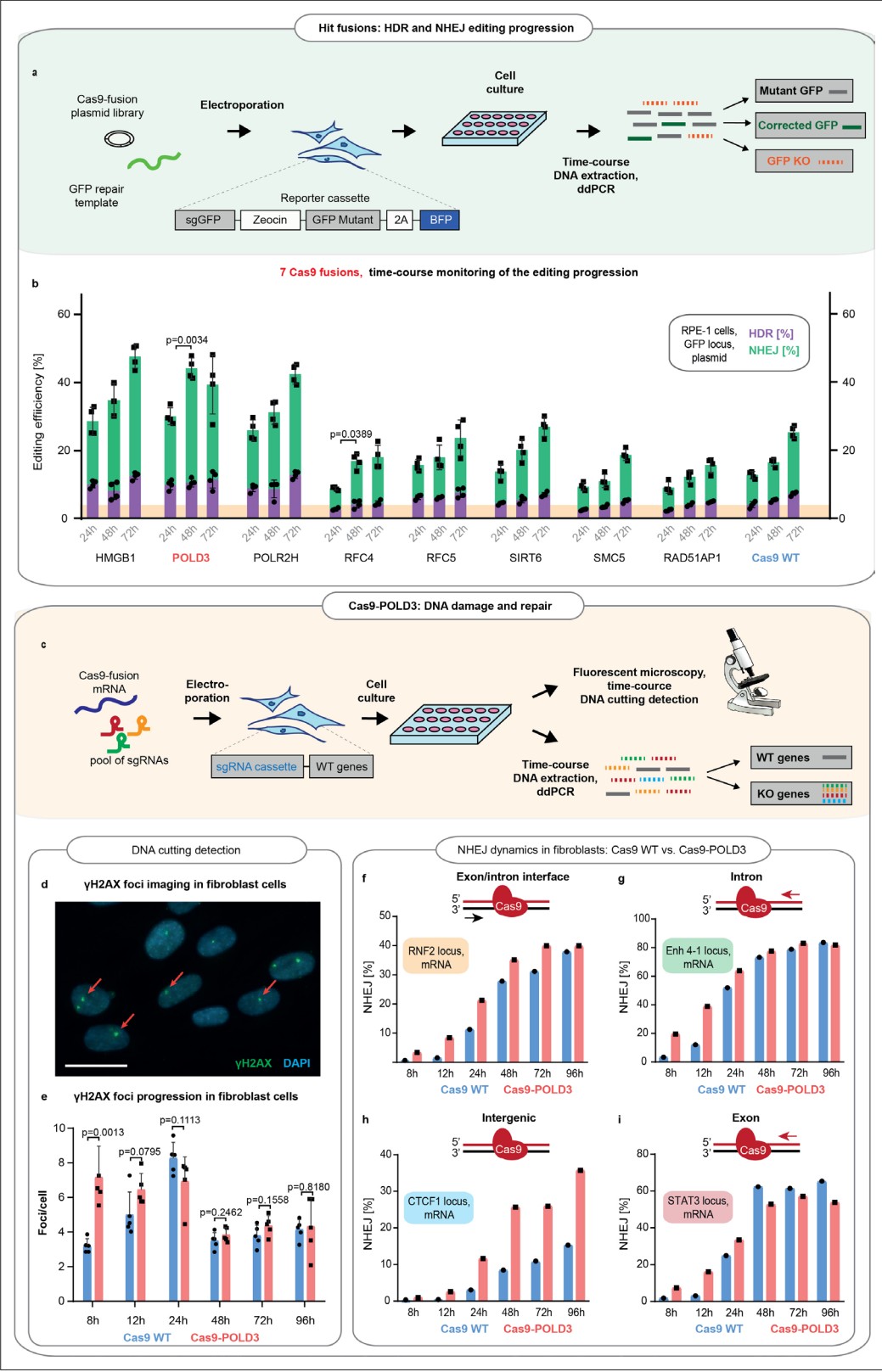

**Figure 3.** Comparison of the DNA repair dynamics between Cas9 fusions. (**a**) Schematic representation of experiment shown in b. (**b**) The DNA repair progression in the retinal pigment epithelium (RPE-1) reporter locus for eight of the better-performing fusions (selected from screens on *Figure 1c-f*). Droplet digital PCR (ddPCR) quantification, n = 3, single independent experiment, bar denotes mean value, error bars represent ± SD. Orange

*Figure 3 continued on next page*

*Figure 3 continued*

highlighting indicates the Cas9WT homology-directed repair (HDR) editing level at 24 hr post-electroporation. p-Values denote significance of the total editing (HDR+ non-homologous end-joining [NHEJ]) increment between the Cas9WT and other fusions (for 24–48 hr period). Statistical values derived using one-way ANOVA test. Cas9 was delivered as plasmid, the repair template as single-stranded oligonucleotide, and the guide was expressed from the genome. (**c**) Schematic representation of experiment shown in **d–i**. (**d**) Representative immunofluorescence image used for DNA breakpoint quantification. Cell nuclei are depicted in blue, γH2AX foci in green, and red arrows indicate individual foci. Scale bar 50 µm. (**e**) Time course for γH2AX foci emergence in human immortalized fibroblasts (BJ-5ta). The cells were electroporated with Cas9WT or Cas9 fusion to DNA polymerase delta subunit 3 (Cas9-POLD3) mRNA and a pool of three guides targeting RNF2, Enh 4–1, and STAT3 loci. The CTCF1 gRNA is constitutively expressed. Five confocal images were taken for each condition, n = 5, single independent experiment, bar denotes mean value, error bars represent ± SD. Statistical significance of the difference between Cas9WT and Cas9-POLD3 is calculated using ANOVA test for the equality of the means at a particular time point. (**f–i**) ddPCR quantification of the NHEJ repair dynamics across time for the experiment described in (**e**). The red arrow shows the direction of the approaching polymerase. The DNA strand which the CRISPR-Cas9 binds to is colored in red. n = 1, single independent experiment, bar denotes mean value, error bars represent ± SD. Cas9 was delivered as mRNA, the repair template as single-stranded oligonucleotide, one guide was expressed from the genome and three guides transfected as synthesized oligonucleotides.

locus (*Figure 3f-i*). We saw a profoundly increased NHEJ in Cas9-POLD3 treated samples in early (8–12 hr) time points, with the exact change depending on the locus and becoming less distinct after 24 hr. All these experiments suggest that the mechanism for improved gene editing for Cas9-POLD3 is the rapid initiation of DNA repair, possibly due to the early recruitment and removal of the fusion from the DNA. The effect size is locus dependent and is not evident in other fusions where the editing follows the kinetics of Cas9WT.

## Cas9-POLD3 interacts with ATP-dependent helicases

To better understand how the fusion proteins modify editing, we looked at their formed molecular interactions (interactomes) using affinity purification mass spectrometry (AP-MS) (*Figure 4a, b*, *Supplementary file 2*). We first constructed HEK293T cell lines consisting of the GFP reporter and tet-inducible, MAC-tagged (*Liu et al., 2018*) Cas9 fusions in the isogenic flip-in locus (*Figure 4—figure supplement 1*). When Cas9 expression is induced, it complexes with the GFP targeting guide and binds and cuts the GFP sequence. The MAC tag in the Cas9 ribonucleoprotein (RNP) complex enables the affinity purification of the complex. Additionally, the MAC tag enables proximity-dependent biotinylation of the interacting and close proximity proteins that can be subsequently purified and identified by mass spectrometry (BioID labelling).

We mapped the stable molecular interactions for Cas9WT and five Cas9 fusions. These include three DNA replication fork proteins (POLD3, RFC4, and RFC5), the RNA polymerase POLR2H and HMGB1, which is a structural chromatin component and a nucleosome remodeler. We recovered a total of 76 Cas9-POLD3 interactions, none of which have been previously reported (there are a total of nine reported POLD3 interactions). We did not recover any polymerase δ subunit components, likely because the fusion uses the short POLD3 isoform 3 (ENST00000532497.5) which lacks the amino acids (AAs) 1–106 that interact with other polymerase δ subunits (*Lancey et al., 2020*). Cas9-RFC4 and Cas9-RFC5 had one known interactor for each (both RFC4 and RFC5 have >40 reported interactions). Cas9-HMGB1 had 103 interactors (29 previously reported for HMGB1). The landscape for the Cas9 fusions thus differs somewhat from their corresponding endogenous proteins, particularly for the replication fork components.

Cas9WT and all of the fusions recovered a large number of RNA-binding proteins that presumably bind to the guide, as well as histones and other structural DNA components. The unique interactions of the Cas9 fusions included proteins with known nucleosome remodeling and helicase activity (*Supplementary file 3*). These include the minichromosome maintenance protein complex (MCM) that interacts with POLD3. Other helicases were RUVB1 and -2 that interacted with the other fusions. MCM and RUVB1-2 are helicase complexes that participate in DNA repair and replication fork stability maintenance (*Rajendra et al., 2014*; *Lossaint et al., 2013*). The helicases might aid editing by opening DNA and making it more accessible to Cas9 binding. After cutting, they might dislodge Cas9 from

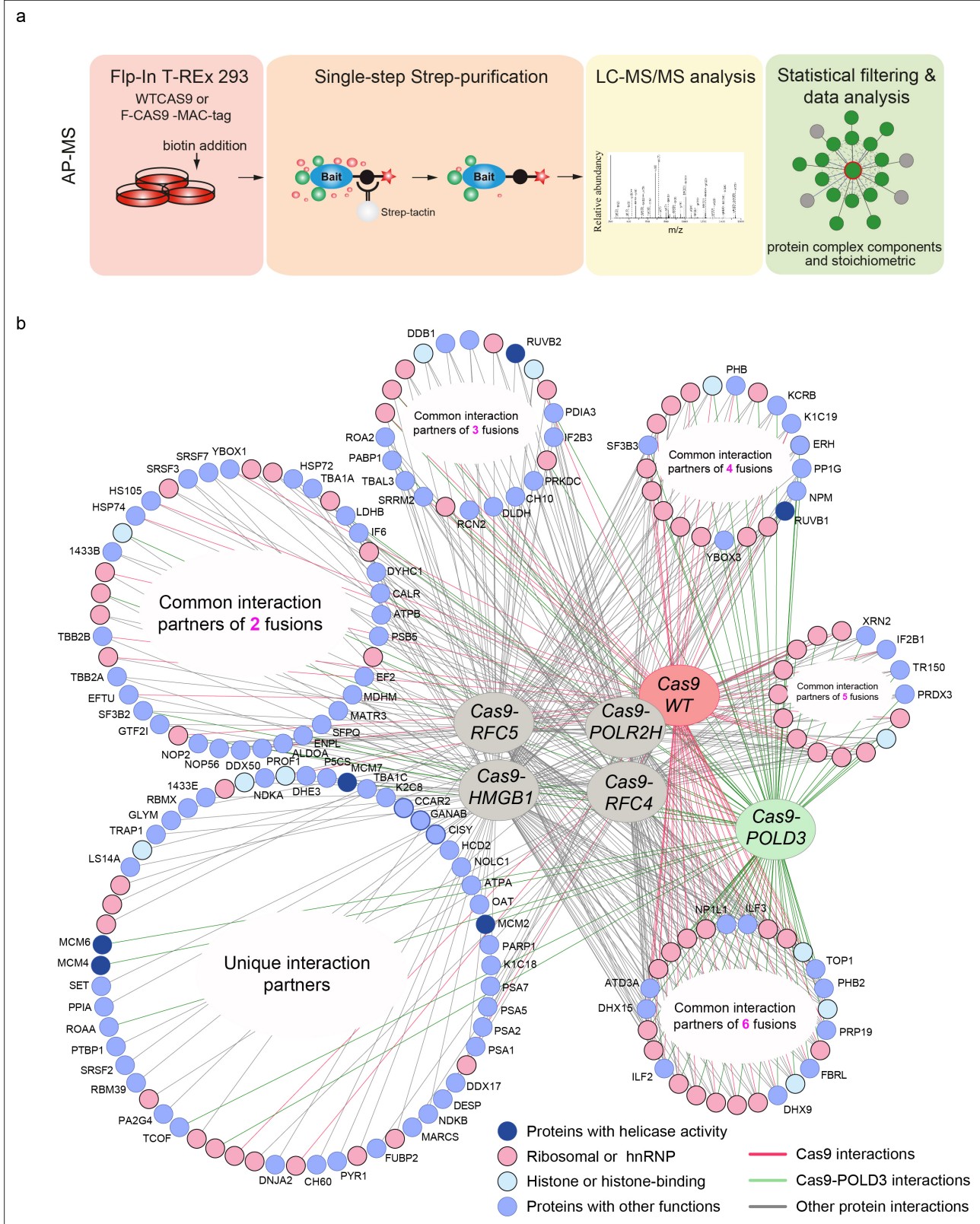

**Figure 4.** Protein-protein interactions of editing-enhancing Cas9 fusions. (**a**) Schematic representation of the affinity purification mass spectrometry (AP-MS) experimental workflow. (**b**) Protein interaction map of the Cas9 fusions and Cas9WT. The interacting endogenous proteins are clustered into circles based on how many Cas9 proteins they interact with. The endogenous proteins in the bottom left circle interact with only one fusion or Cas9WT, and the proteins in the bottom right circle interact with all the fusions or Cas9WT. Pink coloring represents RNA-binding proteins, light-blue represents

*Figure 4 continued on next page*

*Figure 4 continued*

histone proteins, dark blue indicates interaction partners with helicase activity, and light blue circles denote proteins with other functions. Each AP-MS experiment was conducted in three biological replicates (n = 3). The Cas9 and gRNA were expressed from the genome.

The online version of this article includes the following figure supplement(s) for figure 4:

**Figure supplement 1.** Protein expression quality control for mass spectrometry analysis.

the cutting site and make the DNA ends available for processing by the DNA repair machinery (*Richardson et al., 2016*, *Jones et al., 2017*, *Clarke et al., 2018*).

## Cas9-POLD3 fusion does not increase off-target cuts

The POLD3 fusion alters the DNA-binding kinetics of the CRISPR complex and might affect the specificity of the CRISPR binding and cutting. Therefore, we compared the off-target profiles of the Cas9WT and Cas9-POLD3 using GUIDE-Seq (*Tsai et al., 2015*). Briefly, we electroporated HEK293T cells with a plasmid encoding the fusion, a plasmid encoding the guide, and blunt phosphorylated double-stranded oligodeoxynucleotides (dsODN), which incorporate in the DNA double-strand breaks and are used for selective amplification and sequencing of the DNA break sites. We used a previously published guide (*Tsai et al., 2015*) against HEK site 4, which is a non-coding target in the HEK293T genome. The cutting with this guide leads to a large number of off-target events, with several off-target sequences being cut more frequently than the on-target site (*Tsai et al., 2015*). The off-target signature of the Cas9-POLD3 was largely overlapping with the WT nuclease, with slightly less cut sites for Cas9-POLD3 (*Figure 5a, b*). Based on this data, it is unlikely that the polymerase fusion affects the cutting specificity of Cas9.

CRISPR-Cas9 gene editing triggers p53-mediated DNA damage response (*Haapaniemi et al., 2018*) that is proportional to the degree of DNA cutting (*Schiroli et al., 2019*). To quantify p53 activation, we analyzed the p21 (CDKN1A) expression from RPE-1 cells edited with Cas9WT and Cas9-POLD3 plasmids and an RNF2-targeting guide with no reported off-target effects (*Tsai et al., 2015*). We noted no difference in p53 activation between the treatment groups (*Figure 5—figure supplement 1*), suggesting that Cas9-POLD3 does not trigger an overt or long-lasting DNA damage response.

Finally, we compared the indel profiles of Cas9WT and Cas9-POLD3 by deep sequencing of the RNF2 and ELANE target loci (*Figure 5c-f*). We used unique molecular identifiers (UMIs) in the amplicon primers to account for potential PCR bias. The indel profiles were dependent on the target sequence, with NHEJ (1 bp deletion) dominating in RNF2 and MMEJ (23 bp deletion) dominating in ELANE locus (*Figure 5g, h*). The outcomes were generally similar between Cas9WT and Cas9-POLD3. Based on these experiments, the polymerase fusion does not affect the CRISPR on- or off-target repair outcomes. However, a larger experiment is necessary for a more comprehensive map of the specificity and the repair outcomes for Cas9-POLD3.

## Cas9-POLD3 fusion and the previously reported editing-enhancing fusions

Several Cas9 fusions have previously been reported to increase HDR editing (*Charpentier et al., 2018*, *Jayavaradhan et al., 2019*, *Yang et al., 2018*). Of the published fusions, our screening independently identified Cas9-HMGB1 (*Ding et al., 2019*) as an editing enhancer. For a broader comparison, we evaluated the performance of Cas9-POLD3 against five additional published fusions: Cas9-DN1S (fragment inhibiting NHEJ initiation by 53BP1 blockade) (*Jayavaradhan et al., 2019*), Cas9-CtIP (a protein that promotes HDR pathway choice) (*Charpentier et al., 2018*), Cas9-HE (active fragment of CtIP) (*Charpentier et al., 2018*), Cas9-Geminin (fragment that targets the Cas9 expression to S-G2 phases of the cell cycle) (*Yang et al., 2018*), and Cas9 HE-Geminin (the fusion combines both HE and Geminin fragments) (*Charpentier et al., 2018*). The five fusions showed comparable performance with Cas9-POLD3 in GFP locus in our model cell line, with S cell cycle phase targeting providing the highest benefit (*Figure 6a*). We further tested four fusions in four endogenous loci in BJ-5ta, RPE-1 and HEK293T lines using two delivery modes (plasmid and mRNA) (*Figure 6b* and *Figure 6—figure supplement 1*). The editing enhancement was locus- and cell-type dependent, in line with our observations from the larger screens.

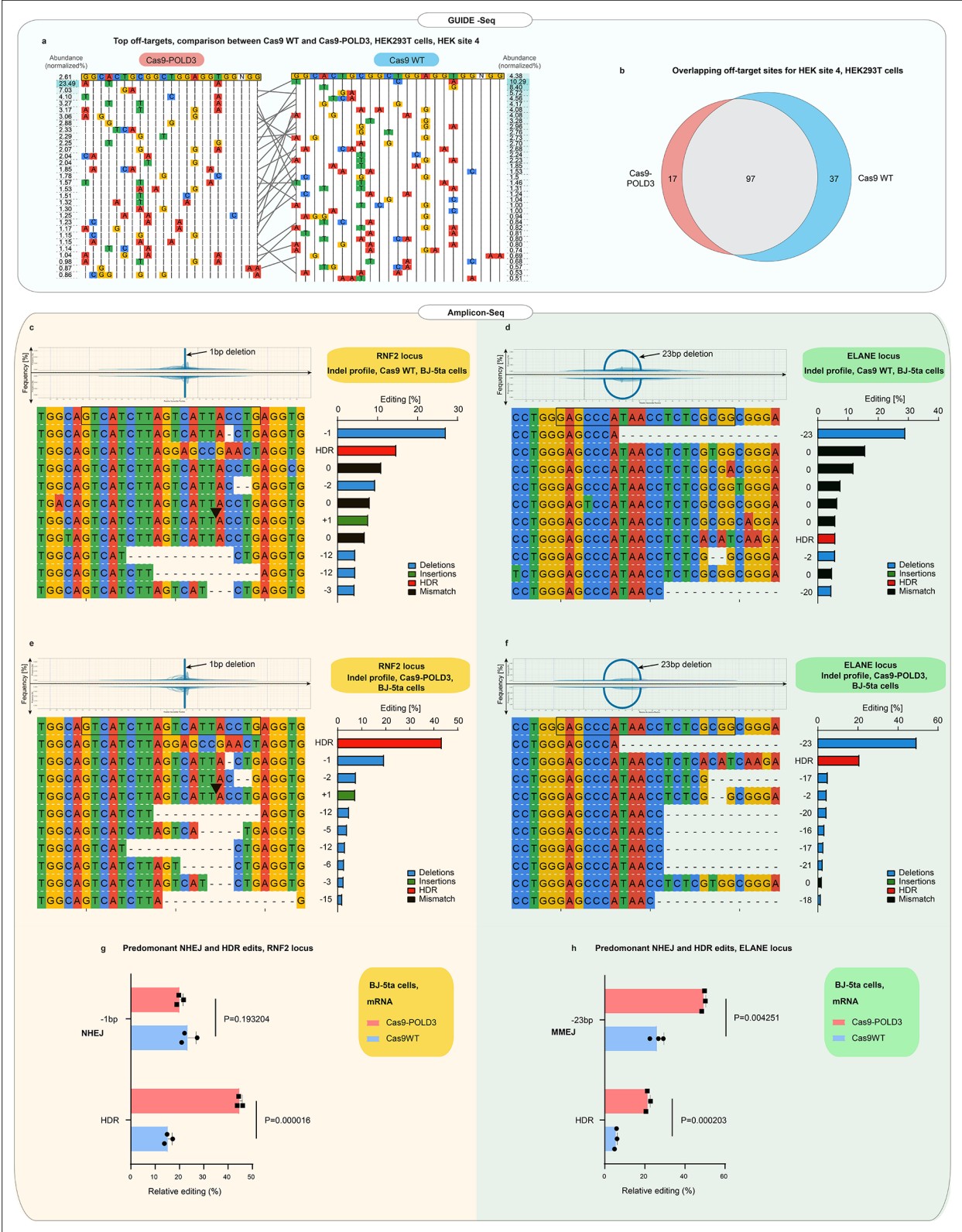

**Figure 5.** Off-target and indel profiles of wild-type (WT) and Cas9 fusion to DNA polymerase delta subunit 3 (Cas9-POLD3) nucleases. (**a**) Mismatch plot of the GUIDE-Seq (*Tsai et al., 2015*) results for Cas9WT and Cas9-POLD3, using a published guide (*Tsai et al., 2015*) against the endogenous human embryonic kidney (HEK) site 4. Cas9 and the guide were co-transfected as plasmids. The on-target sequence is depicted at the first line of the table. The most abundant off-targets are listed underneath the intended edit and sorted by frequency, which is determined by dividing the number of reads

*Figure 5 continued on next page*

*Figure 5 continued*

that contain the off-target edit with the total read count. Gray lines connect the shared off-target sites. (**b**) Venn diagram of the common and unique off-target sites between Cas9WT and Cas9-POLD3. (**c–f**) Mismatch plots of the indel profiles of Cas9WT and Cas9-POLD3 in BJ-5ta fibroblasts, obtained by deep amplicon sequencing. Cas9 was delivered as mRNA, the repair template as single-stranded oligonucleotide, and the guide was expressed from the genome. The on-target gRNA-binding site is on the top row, highlighted with a black box. The most frequent indels are below, sorted according to their frequency. The indel frequency is calculated by dividing the indel read count by the sum of the top 10 indel read counts. The edits matching the homology-directed repair (HDR) template are marked in red in the summary plots. Plots depict: (**c**) On-target editing signature of Cas9WT, RNF2 locus. (**d**) On-target editing signature of Cas9WT, ELANE locus. (**e**) On-target editing signature of Cas9-POLD3, RNF2 locus. (**f**) On-target editing signature of Cas9-POLD3, ELANE locus. (**g–h**) The most common non-homologous end-joining (NHEJ) and HDR editing outcomes depicted in bar graphs. n = 3, bar denotes mean value, error bars represent ± SD. Statistical significance is calculated with unpaired, two-sided Student's t-test.

The online version of this article includes the following figure supplement(s) for figure 5:

**Figure supplement 1.** P21 expression upon CRISPR-Cas9-POLD3 editing.

To understand whether it was possible to use the fusions as CRISPR-Cas9 RNP complexes, we produced Cas9 fusions to POLD3, HMGB1, and HMGB2 in *Escherichia coli* and transfected them with cationic lipid transfer to RPE-1 and HEK293T reporter cell lines. We excluded Cas9-Geminin, as it has previously been tested as an RNP complex (*Gutschner et al., 2016*). Cas9-POLD3 performed best in both cell lines, although we cannot exclude differences in transfection efficiency and fusion protein folding to affect editing outcomes (*Figure 6c*). We also tested the effect of Cas9-POLD3 fusion activity in human embryonic stem cells (hESCs) line and stimulated peripheral blood mononuclear cells (PBMCs) (*Figure 6—figure supplement 2*). Although the POLD3 fusion showed a trend for improvement in both cell types, the benefit is less pronounced than in the model RPE-1 and fibroblast cells possibly due to cell-type-specific differences in cell proliferation rate and DNA damage and repair signaling (*Miyaoka et al., 2016*).

## Discussion

Through screening 450 human proteins involved in DNA repair, we have identified a subset that can improve genome editing as Cas9WT fusions. The editing-enhancing fusions show locus- and cell-type-specific effects. We identify Cas9 fusion to POLD3 as a novel editing enhancer and propose the mechanism of action to be the rapid initiation of DNA repair.

DNA polymerase delta subunit 3 (POLD3) is a component of the replicative polymerase δ, an enzyme that is involved in DNA replication and repair (*Slade, 2018*, *Tumini et al., 2016*, *Tan et al., 2020*). We show that the POLD3 fusion to Cas9 accelerates the initiation of the CRISPR cut repair. The CRISPR complex binds to DNA for 6–8 hr after cutting and needs active removal from the DNA prior to the start of the repair. Approaching RNA polymerase can dislodge Cas9 from the DNA break, which exposes the cut strands to the DNA repair machinery and speeds up the repair process (*Clarke et al., 2018*). We hypothesize that POLD3 fusion partially shares this mechanism and improves genome editing by accelerating the removal of Cas9 from the cut site (*Clarke et al., 2018*). The faster Cas9 removal results in early recruitment of the DNA damage response machinery to the break, which speeds up the DNA repair progression. Cas9-POLD3 performs best in BJ-5ta cells, presumably due to their slow proliferation. Rapid cell doubling rate will lead to more active physiological polymerase engagement along the DNA, which in turn will naturally hasten the removal of the Cas9 nuclease.

Cas9 fusions can stimulate homologous recombination in a number of ways, including altering the local balance of DNA repair factors (*Jayavaradhan et al., 2019*, *Charpentier et al., 2018*), timing the editing to S cell cycle phase (*Yang et al., 2018*), or improving chromatin accessibility (*Ding et al., 2019*). POLD3 fusion enhances editing by speeding up the initiation of DNA repair, and this effect might increase by using the individual domains of the POLD3 protein. In comparison to the previously reported fusions, POLD3 fusion provides a novel mechanism for editing enhancement, and combining POLD3 to protein domains with different editing-enhancing mechanisms might lead to additive benefit. Rapid and efficient gene editing would be particularly useful when editing primary stem cells, which cannot be sustained for a long time in vitro, or in medical applications which require high precision and utilize diseased patient cells that are sensitive to generalized alterations in DNA damage signaling and chromatin remodeling.

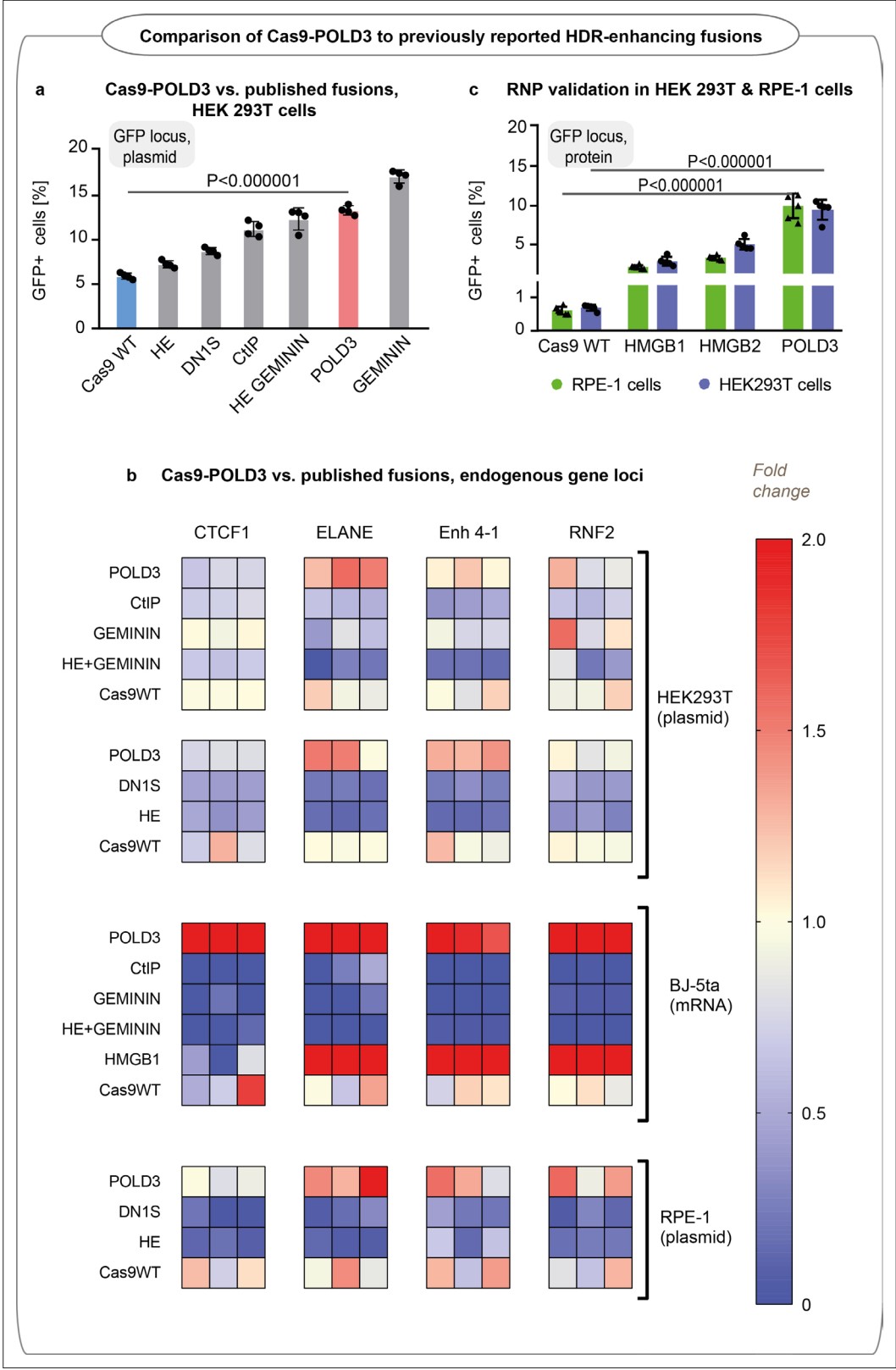

**Figure 6.** Editing efficiency of Cas9 fusion to DNA polymerase delta subunit 3 (Cas9-POLD3) and a panel of various homology-directed repair (HDR) improving fusions reported in the literature. (**a**) Editing of GFP in reporter human embryonic kidney (HEK293T) cells, n = 4, one of two independent experiments, bar denotes mean value, error bars represent ± SD. Statistical significance is calculated with unpaired, two-sided Student's t-test. Cas9 was

*Figure 6 continued on next page*

*Figure 6 continued*

delivered as plasmid, the repair template as single-stranded oligonucleotide, and the guide was expressed from the genome. (**b**) Normalized to Cas9WT HDR efficiency of the fusions in four endogenous loci (CTCF1, ELANE, Enh 4–1, RNF2) in HEK293T, hTERT immortalized fibroblasts (BJ-5ta) and RPE-1 cells quantified by droplet digital PCR (ddPCR). Cas9 was delivered as plasmid (HEK293T and RPE-1) or mRNA (BJ-5ta), the repair template as single-stranded oligonucleotide, and the guide was added in electroporation reaction (HEK293T and RPE-1) or expressed from the genome (BJ-5ta). Heat maps represent HDR values normalized to experimental average for each gene locus. Each cell type was tested in a single biological experiment (n = 3 HEK293T [top dataset] and BJ-5ta; n = 4 HEK293T [bottom dataset] and RPE-1). (**c**) GFP reporter locus editing efficiency of recombinant Cas9 fusion proteins in RPE-1 and HEK293T cells. n = 5, one of two independent experiments, bar denotes mean value, error bars represent ± SD. Statistical significance is calculated with ANOVA that is adjusted for multiple comparisons.

The online version of this article includes the following figure supplement(s) for figure 6:

**Figure supplement 1.** Comparison of Cas9-POLD3 fusions to previously published fusions in human embryonic kidney (HEK293T) cells, immortalized fibroblasts (BJ-5ta), and retinal pigment epithelium (RPE-1) cells.

**Figure supplement 2.** Cas9 fusion to DNA polymerase delta subunit 3 (Cas9-POLD3) editing performance in human embryonic stem cells (hESCs) and peripheral blood mononuclear cells (PBMCs).

The efficiency of CRISPR cutting and the subsequent DNA repair is dependent on the individual locus, where the cell type (*Miyaoka et al., 2016*), guide sequence (*Gisler et al., 2019*), chromatin state (*Verkuijl and Rots, 2019*), the DNA strand orientation (*Clarke et al., 2018*), and the level of the locus transcription (*Verkuijl and Rots, 2019*) all affect the editing outcome. In combination, these factors contribute to the locus- and cell-type-specific editing enhancement that we observe in all of the studied fusions. In addition to altering the access of the endogenous proteins to the DNA break, the Cas9 fusion likely interferes with the endogenous Cas9 activity, and altering the linker sequence between the Cas9 and the fusion protein might further affect the editing efficiency. Our findings underscore the importance of studying new editing-enhancing strategies directly in the end-user applications, as it can be challenging to find CRISPR-Cas9 fusions that promote HDR across diverse loci in a variety of cell types. The overall editing enhancement for all the Cas9 fusions was up to twofold (~20% total HDR) in the models used in this study, and it remains to be seen how this effect shows in fully optimized conditions.

In conclusion, we have extensively compared the known human DNA repair proteins and protein domains for their ability to locally improve CRISPR-Cas9 genome editing. We confirm that rapid removal of the CRISPR complex from the cut site and improved chromatin accessibility can locally promote HDR outcomes. We also note that the effects are locus- and cell-type-specific, suggesting that the strategy needs to be tailored to individual applications with consideration of cell type, locus, gRNA and other factors.

# Materials and methods
## Cell lines
hTERT immortalized retinal pigment epithelium cells were purchased from ATCC. HEK293T cells and BJ-5ta cells (to prepare lentiviral constructs for crispr screens) were purchased from ATCC. The hESC H13 line was obtained from the WiCell Research Institute (Madison, WI). Cell lines were purchased directly from the vendor. ATCC uses short tandem repeat profiling to confirm identity of the cell lines. In addition, the cell morphology was consistent with RPE-1, HEK293T, BJ-5ta, and hESC morphology. All cell lines tested negative for mycoplasma during the time of the study. We used Mycoplasmacheck service from Eurofins Genomics. Cell samples were prepared according to the company instructions, labeled with individual barcodes and sent for the testing. No misidentified cell lines were used in the study.

## Cell culture
HEK293T, RPE-1, and BJ-5ta cells were cultured at 37°C in a humidified incubator in DMEM (Thermo Fisher Scientific) supplemented with 10% fetal bovine serum (FBS; Thermo Fisher Scientific) and 1% penicillin-streptomycin (Thermo Fisher Scientific). After electroporation, cells were cultured in DMEM (Thermo Fisher Scientific) supplemented with 10% FBS (Thermo Fisher Scientific), but without

penicillin-streptomycin. Cells were split routinely twice a week using TrypLE Express Enzyme (Thermo Fisher Scientific).

## RPE-1 and HEK293T GFP reporter cell lines

For easy detection of CRISPR-Cas9-mediated homologous recombination, we created a cassette encoding (1) Zeocin resistance gene for selecting stable integrants, (2) mutant GFP (37A-38A-39A) sequence, (3) BFP or red fluorescent protein (RFP) sequence separated from GFP by a 2A self-cleaving peptide, and (4) sgRNA targeting the mutant GFP. When Cas9 along with a repair template correcting the GFP mutation is introduced into these cell lines, gene correction will give rise to functional GFP. GFP fluorescence was measured by fluorescence-activated cell sorting (FACS) after 5 days if Cas9 was delivered as a plasmid, or 4 days if Cas9 was delivered as RNP.

The reporter cassette sequence was cloned to pLenti sgRNA(MS2) plasmid backbone (Addgene #61427). The cassette sequence is shown in *Supplementary file 8* and the plasmid will be deposited to Addgene.

## Lentivirus production and transduction

The reporter cassette was packaged into lentiviral particles by transfecting HEK293T cells with the reporter cassette plasmid and the two packaging plasmids psPAX2 (Addgene #12260) and pCMV-VSV-G (Addgene # 8454) in equimolar ratios. After 48 hr, the virus-containing supernatant was concentrated 40-fold using Lenti-X concentrator (Clontech). Single-use aliquots were prepared and stored at –140°C.

The cassette was transduced to HEK293T and RPE-1 cells by lentivirus at low multiplicity of infection. The cells were propagated in Zeocin (500 µg/ml) selection media for 2 weeks. After selection, single-cell clones were obtained by sorting BFP+ cells to 96-well plates by flow cytometry. The clones were expanded and frozen for future use.

## BJ-5ta and HEK293T sgRNA cell lines

The cell lines express a guide targeting one of the six endogenous loci: RNF2, ELANE, Enh1-4, STAT3, FANCF. The CRISPR RNA sequences (shown in *Supplementary file 5*) were synthesized by Eurofins Genomics and cloned to pLentiPuro plasmid (Addgene #52963) according to published instructions (*Sanjana et al., 2014*). The plasmid identities were verified by Sanger sequencing.

The lentiviruses were prepared and transduced to HEK293T and BJ-5ta cells as described above. The cells were selected with 10 µg/ml puromycin (Thermo Fisher Scientific) or 10 µg/ml blasticidin (Sigma Aldrich) for 7 days prior to aliquoting and freezing.

## DNA repair protein library

The DNA repair protein open reading frames (ORFs) and their corresponding sequences are found in the *Supplementary file 1* . In addition, the screen set contains 21 prokaryotic exonucleases as well as four engineered protein tags that are inert and function comparably to WTCas9. The corresponding full-lenght ORFs were either picked from the human ORFeome (hORF8.1) or synthesized by Genscript Inc Since Cas9 is a large protein of >2000 AAs, we expected the largest DNA repair proteins (>1000 AA) to lead to a failure in fusion protein expression. Therefore, the DNA repair proteins that exceeded the size of 600 AAs were processed into smaller fragments (100-500 AAs) in between the domain boundaries. The fragments were synthesized by Genscript Inc.

## Expression plasmids

By utilizing the commercial pcDNA-DEST40 backbone (Invitrogen), we constructed a custom mammalian expression plasmid containing a WTCas9 sequence with N- and C-terminal nuclear localization signals, followed by a Gateway recombination site (referred to as Cas9-GW plasmid). Genscript Inc performed the insert synthesis and cloning to pcDNA-DEST40 backbone.

## High-throughput gateway cloning and plasmid preparation

The DNA repair protein inserts were cloned to Cas9-GW plasmids in 96-well plates in 5 l reactions, with 1 µl Gateway LR clonase II enzyme mix (Invitrogen), 50 ng entry plasmid, and 100 ng destination plasmids, respectively, and Tris-EDTA (TE) buffer (pH 8) added to a final volume of 5 µl. After

overnight incubation, we transformed 1 µl of the reaction mix to 15–20 µl of DH5-α competent cells (Invitrogen). Plasmids were extracted in 96-well format using the Wizard SV 96 plasmid miniprep kit (Promega) according to manufacturer's instructions using the Biomek FXP liquid handling system (Beckman Coulter), and eluted to 100 µl TE buffer (pH 8). This yielded a destination plasmid concentration of ~150 ng in the majority of the wells.

Of the resulting five 96-well destination clone plates, we verified one plate by Sanger sequencing. After each screening round, the top hits were validated by Sanger sequencing.

## Cell transfections

HEK293T cells containing the GFP-BFP-sgRNA reporter cassette were split into 24-well plates at ~100,000 cells/well, 500 µl final volume. The next day, we replaced the media and prepared the transfection complexes in 96-well plates using Fugene HD (Promega) with the following modifications: with 25 µl Opti-MEM (Thermo Fisher), 2 µl Fugene HD (Promega), 400 ng plasmid, and 12.5 pmol repair DNA (synthesized by Eurofins Genomics). After 20 min incubation, the cells were transfected in triplicate, with the end volume of 25 µl transfection mix in each well. Transfections were conducted with Biomek FXP liquid handling system (Beckman Coulter). The cells were grown for 5 days and GFP expression evaluated by FACS as described in the corresponding section.

## Fluorescence-activated cell sorting

For the detection of corrected GFP, the Cas9-transfected cells were cultured for 5 days, trypsinized, resuspended in warm culture medium and immediately subjected to FACS. The data was acquired by CyAn II flow cytometer (Beckman Coulter) coupled with Hypercyte robotics (Intellicyt) or iQue screener plus (Intellicyt).

Well information was extracted from the plate fcs file using Kaluza v.1.2 or 2.1 (Beckman Coulter). The number of GFP+ cells per well was obtained by batch gating. The gates used in analyzing the screen FACS data are shown in *Supplementary file 6*. The effect size for each Cas9 fusion was calculated by normalizing the GFP expression of the individual well either to the average GFP expression of all wells in the plate or to the overall experimental average. The statistical significance was calculated using standard one-way ANOVA test (*Supplementary file 4*).

## Cas9WT fusion protein validation in HEK293T and RPE-1 reporter cells

We chose 52 Cas9WT-DNA repair fusion constructs for validation. We picked the constructs from the original plates using Biomek i7 robotics and prepared new minipreps from the original constructs using the Wizard SV 96 Plasmid DNA Purification System (Promega), according to the manufacturer's instructions. The constructs were Sanger sequenced to validate construct identity.

HEK293T cells containing the GFP-RFP-sgRNA reporter cassette were transfected as described in the corresponding section, with the following modifications: seeding on the 48-well plate, 30,000 cells/well; transfection mix: 12.5 µl Opti-MEM (Thermo Fisher), 1.25 µl Fugene HD (Promega), 250 ng plasmid, and 6 pmol repair DNA (synthesized by Eurofins Genomics). After 20 min incubation, the cells were transfected in four replicas. The transfections were performed in parallel plates to account for batch variation between cell culture plates. Transfections were conducted with Biomek i7 liquid handling system (Beckman Coulter). The cells were grown for 5 days and GFP expression evaluated by FACS.

RPE cells containing the GFP-RFP-sgRNA or GFP-BFP-sgRNA reporter cassette were electroporated with Lonza 4D 96-well electroporation system; 400 ng of the plasmid was pre-mixed with 40 pmol of the repair template on the 96-well PCR plate. RPE cells were trypsinized, washed with PBS, and resuspended in the 1 M electroporation buffer (5 mM KCl, 15 mM MgCl$_2$, 120 mM Na$_2$HPO$_4$/NaH$_2$PO$_4$, pH 7.2, and 50 mM mannitol) to obtain a cell density of 200,000 cells per 20 µl of the buffer (ratio is for single-well reaction in the electroporation plate); Twenty µl of the cell suspension was then dispensed in each well of the 96-well plate containing pre-mixed plasmids and repair template, gently mixed and transferred to the 96-well electroporation plate (Lonza). Electroporation was conducted using the pulse code EA-104. Instantly after electroporation, 80 µl of pre-warmed medium (DMEM, supplemented with 10% FBS, no antibiotics) was added into each well. Cells were collected and plated on 48-well plates (Costar) containing 300 µl of pre-warmed medium (DMEM, supplemented

with 10% FBS, no antibiotics). Liquid handling steps were done with the Biomek i7 platform. Electroporated cells were cultured for 5 days, after which the GFP expression was evaluated by FACS.

## Cas9WT fusion protein validation in endogenous loci (HEK293T cells)

We chose 31 Cas9WT-DNA repair fusion constructs for validation in endogenous loci in HEK293T cells. We used four different HEK293T lines, each stably expressing one of the sgRNAs targeting ELANE, CTCF1, Ench4-1, RNF2 genomic loci (CRISPR RNA sequences in *Supplementary file 5*). Cells seeded to 24-well plates at ~100,000 cells/well were transfected as described above, with the following modifications: 25 µl Opti-MEM (Thermo Fisher), 2.5 µl Fugene HD (Promega), 480 ng of Cas9 fusion plasmid, and 10 pmol repair DNA. After 20 min incubation, the cells were transfected in triplicate. After 5 days of incubation, DNA was extracted with PureLink 96 Genomic DNA kit (Invitrogen) and analyzed using ddPCR.

## Droplet digital PCR gene editing evaluation

We performed ddPCR assays to estimate NHEJ and HDR efficiencies at the RNF2, Enh 4–1, FANCF, ELANE, STAT3, CTCF1, and GFP guide target loci. The assay schematic is shown in *Supplementary file 7*. The fwd and reverse amplicon primers bind outside the homology arms of the repair DNA template. The ddPCR was performed on the QX200 system (Bio-Rad Laboratories). Final reaction mixture volume was 20 µl: 10 µl of 2× ddPCR Supermix for Probes (Bio-Rad Laboratories, 1863024), 8 µl of DNA (concentration normalized to 8 ng/µl), primers (900 nM), reference probe (250 nM), and HDR or NHEJ probe (250 nM). The HDR and NHEJ detection occurs in two separate ddPCR. Each reaction was then loaded into a sample well of an eight-well disposable cartridge (DG8; Bio-Rad Laboratories) along with 70 µl of droplet generation oil (Bio-Rad Laboratories). Droplets were formed using a QX200 Droplet Generator (Bio-Rad Laboratories). Droplets were transferred to a 96-well PCR plate, heat-sealed with foil, and amplified using a conventional thermal cycler. The thermocycling protocol was the following: (1) 95°C - 10 min, (2) 94°C – 30 s, 56°C – 3 min, step repeated 42 times (3) 98°C – 10 min, (4) 4°C – hold. The resulting PCR products were loaded on a QX200 Droplet Reader (Bio-Rad Laboratories), and the data analyzed using QuantaSoft software (Bio-Rad Laboratories). Primer and probe sequences are listed in *Supplementary file 7*. The data was analyzed with QuantaSoft software. Representative gatings are shown in *Supplementary file 7*.

## ddPCR for the evaluation of gene expression levels

RNA was extracted from the frozen cell pellets using the Total RNA Purification Kit (Nordic BioSite, 298–17200) according to manufacturer's instructions, including the DNAse treatment step (Nordic BioSite, 298–25710). cDNA synthesis was accomplished using the iScript Advanced cDNA Synthesis Kit (Bio-Rad, Cat. no. 1725038), according to the manufacturer's instructions. Final reaction mixture volume was 20 µl: 10 µl of 2× ddPCR Supermix for Probes (Bio-Rad Laboratories, 1863024), 5 µl of cDNA (concentration normalized to 0.2 ng/µl), 1 µl of Primer-Probe Mix for reference gene (Bio-Rad, GAPDH/HEX ddPCR assay dHsaCPE5031597), 1 µl of Primer-Probe Mix for gene of interest (Bio-Rad, CDKN1A/FAM ddPCR assay dHsaCPE5052298, or IFNB1/FAM ddPCR assay dHsaCPE5040802), 3 µl of DNAse/RNAse milliQ. Each reaction was then loaded into a sample well of an eight-well disposable cartridge (DG8; Bio-Rad Laboratories) along with 70 µl of droplet generation oil (Bio-Rad Laboratories). Droplets were formed using a QX200 Droplet Generator (Bio-Rad Laboratories). Droplets were transferred to a 96-well PCR plate, heat-sealed with foil, and amplified using a conventional thermal cycler. The thermocycling protocol was the following: (1) 95°C – 10 min, (2) 94°C – 30 s, 55°C – 1 min, step repeated 39 times, (3) 98°C – 10 min, (4) 4°C – hold. The resulting PCR products were loaded on a QX200 Droplet Reader (Bio-Rad Laboratories), and the data analyzed using QuantaSoft software (Bio-Rad Laboratories).

## In vitro mRNA transcription

Cas9WT fusion mRNA was prepared with HiScribe T7 ARCA mRNA Kit with tailing (NEB-bionordika), according to the manufacturer's instructions. Total of 8000 ng stock plasmid was digested with 2 µl FastDigest MssI enzyme in the supplemented restriction-digestion buffer, total reaction volume 20 µl. Incubation was carried out at 37°C overnight. Length of the digested product was confirmed by gel electrophoresis. For the IVT reaction, 1000 ng of the linearized plasmid was mixed with 10 µl of

2xARCA/NTP mix and 2 µl of T7 RNA Polymerase mix, and the reaction was incubated for 30 min at 37°C. Sequentially, 2 µl of DNAse enzyme was added and the mixture was incubated at 37°C for 15 min. Poly(A) tailing step was performed by adding 20 µl of milliQ (RNAse free), 5 µl of 10× PolyA polymerase reaction buffer, and 5 µl of 10× PolyA polymerase directly to the IVT reaction, which was incubation at 37°C for 30 min. mRNA was purified using LiCl solution, as described in the manufacturer's protocol. Aliquots were frozen in –80°C for future use.

## Cas9WT fusion protein validation in endogenous loci (BJ-5ta cells)

We chose seven Cas9WT fusion constructs for validation in endogenous loci in immortalized BJ-5ta fibroblasts. We used five different BJ-5ta lines, each stably expressing one of the sgRNAs (IDT) targeting FANCF, Enh 4–1, STAT3, ELANE, or RNF2 genomic loci (CRISPR RNA sequences in *Supplementary file 7*). The Cas9 fusion constructs were transfected as mRNA; IVT was performed as described above. BJ-5ta-sgRNA cell lines were electoporated with Lonza 4D 96-well electroporation system as described in corresponding section, with following modifications: 1000 ng of mRNA and 100 pmol of the corresponding HDR repair template per well, cell density 500,000 of cells per 20 µl of the electroporation buffer. Cells were pulsed using code CA-137. Cell culture was carried out on six-well plates for 5 days, followed by the DNA isolation (DNeasy Blood & Tissue Kit (250), Qiagen). Editing efficiency was evaluated by ddPCR as described in the corresponding section.

## Cas9-POLD3 validation in PBMCs

PBMCs were isolated from whole blood by density gradient separation method using Lymphoprep (StemCell Technologies, #07851). Obtained cells were cultured in the T cell expansion media (RPMI 1640 Medium, GlutaMAX Supplement, HEPES [Gibco, #72400–021] supplemented with 10% FBS [certified, heat inactivated, Gibco, #10082147], 25 µl/ml anti-human CD3/CD28 [StemCell Technologies, #10970], 200 U/ml IL-2 [Recombinant Human IL-2, Peprotech, #200–02], 5 ng/ml IL-7 [Recombinant Human IL-7, Peprotech, #200–07], 5 ng/ml IL-15 [Recombinant Human IL-15, Peprotech, #200–15] with 1× P/S [Penicillin-Streptomycin-Glutamine 100×, Gibco #10378016]) for 3 days prior to the electroporation at $1 \times 10^6$ cells/ml density.

Electroporation was performed using the Lonza 4D Nucleofector System, Core Unit (Lonza Cat. no. AAF-1002B) with a 96-well Shuttle Device (Lonza Cat. no. AAM-1001S). $1 \times 10^6$ of cells were resuspended in 20 µl of the home-made electroporation buffer (5 mM KCl, 15 mM MgCl$_2$, 120 mM Na$_2$HPO$_4$/NaH$_2$PO$_4$, and 50 mM mannitol, pH 7.2), combined with 1000 ng of nuclease/fusion encoding mRNA, 100 pmol of the corresponding sgRNA and 100 pmol of the repair template (ssODN) and electroporated using EO-115 pulse code. After the electroporation, 80 µl of the recovery media (RPMI 1640 Medium, GlutaMAX Supplement, HEPES [Gibco, #72400–021] supplemented with 10% FBS [certified, heat inactivated, Gibco, #10082147], IL-2 at 500 U/ml, without any antibiotics) was added to each well and cells were incubated for 15 min in the 37°C, 5% CO$_2$. Subsequently, cell suspension from each well was transferred to the 400 µl of the recovery medium pre-dispensed on the 24-well plates. Half of the medium volume from each well was replaced with fresh recovery medium supplemented with 1× P/S (Penicillin-Streptomycin-Glutamine 100×, Gibco #10378016) daily, until the sample collection time point at 96 hr post-electroporation. DNA extraction was done using the DNeasy 96 Blood & Tissue Kit (4) (Qiagen, #69581) according to the kit instructions.

## Cas9-POLD3 validation in hESCs

The hESC H13 line (*Thomson et al., 1998*) was obtained from the WiCell Research Institute (Madison, WI) and cultured on mitomycin-C-treated mouse embryonic fibroblasts (MEFs). HESCs were maintained in DMEM/F-12 (Sigma, #D8900-50L) media containing 15% FBS (Capricorn Scientific, #FBS-ES-HI12A), 5% KnockOut Serum Replacement (Gibco, #10828–028), 1% MEM Non-essential Amino Acids (Gibco, #11140–068), 1% GlutaMAX (Gibco, #35050–038), 1% antibiotic-antimycotic solution (Gibco, #15240–062), 55 µM 2-mercaptoethanol (Gibco, #31350–010), and 5 ng/ml of recombinant human bFGF (Miltenyi Biotec, #130093842). The cells were first passaged 5–6 days prior to the start of the experiment using collagenase type IV (1 mg/ml, Life Technologies, #17104019) and transferred into MEF-coated six-well plates. The media was exchanged 24 hr prior to the electroporation and supplemented with 10 µM Rho Kinase (ROCK)-inhibitor (ATCC, Y-27632).

Immediately before the electroporation, hESC colonies were detached from plates using 1 mg/ml collagenase IV (Gibco, #17104019) and sedimented in 10 ml of PBS, pH 7.4, for 5 min. After brief centrifugation (100× $g$, 1 min) PBS was removed and hESC were disrupted into single cells using 0.25% trypsin/EDTA solution (Gibco, 25200–114) for 3 min at the 37°C, 5% $CO_2$. Trypsin was neutralized with hESC media and the cells were pelleted at 150× $g$ for 5 min. The cell pellet was resuspended in 2 ml PBS, pH 7.4, and cells were counted using Trypan blue on the automated cell counter (Countess). Electroporation was performed using the MaxCyte Expert electroporation system with OC-25 × 3 Electroporation Cuvettes (MaxCyte, #238537). For each reaction, 0.4 × 10⁶ of cells were resuspended in 25 µl of the MaxCyte electroporation buffer (MaxCyte, #EPB1), combined with 1000 ng of Cas9 mRNA, 100 pmol sgRNA (IDT) targeting the RNF2 locus and 100 pmol of the repair DNA template and electroporated using Optimization-8 pulse code. After the electroporation, cells in the electroporation cuvettes were incubated for 15 min in the 37°C, 5% $CO_2$. Then, the cell suspension was transferred to 24-well plates pre-coated with Matrigel (Corning, #734–1440) and supplemented with 1 ml hESC culture media (no antibiotics, supplemented with 10 µM ROCK-inhibitor). The media was exchanged 24 and 72 hr after the electroporation. The DNA was extracted after 96 hr using the DNeasy Blood & Tissue Kit (Qiagen, # 69504) according to the manufacturer's instructions.

## Recombinant protein production

The hit Cas9 fusion proteins were synthesized by GeneArt Inc and cloned to pET301/CT-DEST vector (Invitrogen) or pTH21 vector (*van den Berg et al., 2006*) for *E. coli* expression. The Cas9 sequence as well as the sequence and position of nuclear localization signal and GS linker were similar to the Cas9-GW sequence presented in *Supplementary file 8* . The exact sequences are listed in the file DNA repair protein domain library.xlsx.

All proteins contained C-terminal His-tags and were expressed in *E. coli* BL21 (DE3) T1R pRARE2 at 18°C and purified by the Protein Science Facility (PSF) at Karolinska Institutet, Stockholm. Purification was performed using the HisTrap HP column (GE Healthcare) followed by gel filtration step with a HiLoad 16/60 Superdex 200 (GE Healthcare). Purity of the protein preparation was examined using SDS-PAGE followed by Commassie staining. All purified proteins were concentrated, aliquoted, and stored in a storage buffer (20mM HEPES, 300mM NaCl, 10% glycerol, 2mM TCEP, pH 7.5) at –80°C.

## RNP complex preparation and transfection

We obtained the mGFP-targeting CRISPR and TRACR RNAs from Integrated DNA Technologies (IDT). The CRISPR/TRACR hybridization was performed by incubation of crRNA tracrRNA in equimolar concentrations at 95°C followed by gradual cooling down to room temperature (RT). For RNP complex formation nuclease (protein or fusion) was mixed gRNA in 1:2 molar ratio and incubated 15 min RT. For transfection, reporter HEK293T and RPE-1 cells were plated in 48-well plates at 100,000 cells/well and reverse transfected with 13 pmol of RNP and 6 pmol of repair DNA template using the CRISPRmax transfection reagent (Thermo Scientific) according to the manufacturer's instructions. For the detection of corrected GFP, the transfected cells were cultured for 4 days, trypsinized, resuspended in warm culture medium, and immediately subjected to FACS. The data was acquired by CyAn II flow cytometer (Beckman Coulter) coupled with Hypercyte robotics (Intellicyt) and analyzed with Kaluza v.1.2 (Beckman Coulter).

## Functional comparison between Cas9-POLD3 and previously published fusions

HEK293T cells containing the GFP-BFP-sgRNA reporter cassette were transfected as described in the corresponding section, with the following modifications: seeding on the 48-well plate: 30,000 cells/well; transfection mix: 12.5 µl Opti-MEM (Thermo Fisher), 1.25 µl Fugene HD (Promega), 250 ng plasmid, and 6 pmol repair DNA (synthesized by Eurofins Genomics). After 15 min incubation, the cells were transfected in four replicas. The transfections were performed in parallel plates to account for batch variation between cell culture plates.The cells were grown for 5 days and GFP expression evaluated by FACS.

## DNA breakpoint quantification

We used immortalized fibroblasts (BJ-5ta) expressing a guide that targets the CTCF1 locus. The cells were electroporated as described above with a pool of sgRNAs (IDT) targeting the Enh 4–1, STAT3, and RNF2 loci, 100 pmol RNA per reaction. The purpose of using the pool is to increase the total number of edited alleles per nucleus for better visualization. After the electroporation, cells were plated on six-well plates containing clean glass coverslips and cultured as described above.

Samples were collected for staining at 8, 12, 24, 48, 72, and 96 hr post-electroporation. We used the coverslips for γH2AX foci quantification, and the cells in the surrounding space in the well for NHEJ quantification by ddPCR (described in the corresponding section). For confocal microscopy, each sample slide was washed twice with PBS and fixed with 4% PFA for 15 min at RT, followed by another PBS washing step and storage at +4°C in PBS, pH 7.4 until the sample collection was finished. The samples were then permeabilized (0.1% Triton X-100 in PBS for 45 min RT) and blocked (PBS with 0.1% Triton X-100, 5% BSA, and 5% goat serum for 45 min RT). After blocking, Anti-phospho-H2AX (Ser139) (Millipore, 05-636-25UG) was added in 1:500 ratio, diluted in 0.1% Tween-20, 0.5% BSA, and 0.5% goat serum in PBS, and the samples were incubated overnight at 4°C. The following day, coverslips with samples were washed three times with 0.1% Triton X-100 in PBS with shaking and Goat anti-Mouse IgG (H + L) Cross-Adsorbed Secondary Antibody, Alexa Fluor 488 (Thermo Fisher, A-11001) was added in 1:1000 dilution in 0.1% Triton X-100 in PBS and incubated for 1 hr at RT, protected from light. After three additional washes in PBS, nuclei were stained with DAPI (1 µg/ml in PBS). The slides were mounted using Fluoromount Aqueous Mounting Medium (Sigma Aldrich, F4680).

Widefield fluorescence imaging (Z-stacks) was done using Zeiss AxioObserver Z1, with a Plan Apo 40×/1.4 Oil objective and a Hamamatsu ImagEMX2 EMCCD. Imaging settings (filter set, lamp power, and camera exposure) were kept constant for image collection for all samples. Image analysis was done with batch mode in FIJI (ImageJ) with two main steps: (1) Segmentation of region of interest (ROI) to outline cell nuclei with DAPI channel with the FIJI plugin Auto Threshold (Mean) and watershed algorithm. (2) Counting the γH2AX foci in ROIs identified in Step 1 with the FIJI plugin Find Maxima (prominence = 2000). Average foci/nucleus ratio was calculated for each sample based on the quantified results.

## Cas9WT fusion plasmid titration in RPE-1 reporter cells

The Cas9 fusions that showed a trend for improving over Cas9WT were Sanger sequenced to validate construct identity. We then prepared new minipreps from the original constructs using Qiaprep spin Miniprep kit (Qiagen) according to manufacturer's instructions.

For the time course experiment, RPE-GFP-BFP-gRNA cells were electroporated using the Lonza 4D 96-well electroporation system as described in the corresponding section with the following modifications: 200,000 cells per each well in 24-well plates; 400 ng of plasmid, 100 pmol of GFP repair template; pulse code changed to EA-104. Cells were gradually collected at 24, 48, 72 hr post-electroporation. DNA extracted using DNeasy Blood & Tissue Kit (250) (Qiagen, #69504). Editing efficiency was evaluated by ddPCR as described in the corresponding section.

For the gradient titration, RPE-GFP-BFP-gRNA cells were electroporated with Lonza 4D 16-well electroporation system as described in corresponding section with the following modifications: 200,000 cells per each well in 24-well plates; 15, 30, 60, 120 pmol of plasmid, 40 pmol of GFP repair template; pulse code changed to EA-104. FACS analysis was performed after 5 days by iQue screener plus (Intellicyt) and analyzed using Kaluza v 2.1 software (Beckman Coulter), as described in corresponding section.

## AP-MS and BioID proximity labelling

The hit Cas9 fusion proteins were synthesized and cloned to pDNOR221 Gateway entry vector (Invitrogen) by GeneArt Inc Sequences are listed in *Supplementary file 1*. The inserts were cloned to MAC-Tag-C vector (*Liu et al., 2018*)(PMID29568061, Addgene ID #108077), which adds a C-terminal MAC tag (contains Strep-tag and modified minimal biotin ligase) to the Cas9 fusion.

For generation of the stable cell lines inducibly expressing the MAC-tagged versions of the baits, Flp-In T-REx 293 cell lines (Invitrogen, Life Technologies, R78007) were first transduced with the lentivirus containing the GFP-BFP-sgRNA cassette, as described in corresponding section. The cells were co-transfected with the MAC-tagged expression vector and the pOG44 vector (Invitrogen) using the

Fugene HD transfection reagent (Promega). Two days after transfection, cells were selected in 50 µg/ml streptomycin and hygromycin (100 µg/ml) for 2 weeks.

We tested the presence of the fusion and the functionality of the GFP-BFP-sgRNA reporter cassette by FACS prior to the experiment. The Cas9 expression was induced with 1 µg/ml tetracycline for 24 hr, followed by GFP repair template transfection by RNAiMax transfection reagent (Thermo Fisher Scientific) according to manufacturer's instructions.

For AP-MS experiments, each stable cell line was expanded to 80% confluence in 20 × 150 mm² cell culture plates. 2 × 5 plates were used for AP-MS approach, in which 1 µg/ml tetracycline was added for 30 hr induction, and 2 × 5 plates for BioID approach, in which in addition to tetracycline, 50 µM biotin was added for 30 hr before harvesting. Cells from 5 × 150 mm fully confluent dishes (~5 × 10⁷ cells) were pelleted as one biological sample. Thus, each bait protein has two biological replicates in two different experiments. All samples from one experiment were processed in parallel. Samples were snap-frozen and stored at −80°C. Analyses of the baits were performed as two biological replicates.

For the AP-MS approach, the sample was lysed in 3 ml of lysis buffer 1 (0.5% IGEPAL, 50 mM HEPES, pH 8.0, 150 mM NaCl, 50 mM NaF, 1.5 mM NaVO₃, 5 mM EDTA, supplemented with 0.5 mM PMSF and protease inhibitors; Sigma).

For BioID approach, cell pellet was thawed in 3 ml ice-cold lysis buffer 2 (0.5% IGEPAL, 50 mM HEPES, pH 8.0, 150 mM NaCl, 50 mM NaF, 1.5 mM NaVO₃, 5 mM EDTA, 0.1% SDS, supplemented with 0.5 mM PMSF and protease inhibitors; Sigma). Lysates were sonicated and treated with benzonase.

Cleared lysate was obtained by centrifugation and loaded consecutively on spin columns (Bio-Rad) containing lysis buffer 1, prewashed 200 µl Strep-Tactin beads (IBA, GmbH). The beads were then washed 3 × 1 ml with lysis buffer 1 and 4 × 1 ml with wash buffer (50 mM Tris-HCl, pH 8.0, 150 mM NaCl, 50 mM NaF, 5 mM EDTA). Following the final wash, beads were then resuspended in 2 × 300 µl elution buffer (50 mM Tris-HCl, pH 8.0, 150 mM NaCl, 50 mM NaF, 5 mM EDTA, 0.5 mM biotin) for 5 min and eluates collected into Eppendorf tubes, followed by a reduction of the cysteine bonds with 5 mM tris(2-carboxyethyl)phosphine (TCEP) for 30 min at 37°C and alkylation with 10 mM iodoacetamide. The proteins were then digested to peptides with sequencing grade-modified trypsin (Promega, V5113) at 37°C overnight. After quenching with 10% TFA, the samples were desalted by C18 reversed-phase spin columns according to the manufacturer's instructions (Harvard Apparatus). The eluted peptide sample was dried in vacuum centrifuge and reconstituted to a final volume of 30 µl in 0.1% TFA and 1% CH₃CN.

Analysis was performed on a Q-Exactive mass spectrometer using Xcalibur version 3.0.63 coupled with an EASY-nLC 1000 system via an electrospray ionization sprayer (Thermo Fisher Scientific). In detail, peptides were eluted and separated with a C18 precolumn (Acclaim PepMap 100, 75 µm × 2 cm, 3 µm, 100 Å, Thermo Scientific) and analytical column (Acclaim PepMap RSLC, 75 µm × 15 cm, 2 µm, 100 Å; Thermo Scientific), using a 60 min buffer gradient ranging from 5% to 35% buffer B, followed by a 5 min gradient from 35% to 80% buffer B and 10 min gradient from 80% to 100% buffer B at a flow rate of 300 nl/min (buffer A: 0.1% formic acid in 98% HPLC grade water and 2% acetonitrile; buffer B: 0.1% formic acid in 98% acetonitrile and 2% water). For direct LC-MS analysis, 4 µl peptide samples were automatically loaded from an enclosed cooled autosampler. Data-dependent FTMS acquisition was in positive ion mode for 80 min. A full scan (200–2000 m/z) was performed with a resolution of 70,000 followed by top 10 CID-MS2 ion trap scans with resolution of 17,500. Dynamic exclusion was set for 30 s.

Acquired MS2 spectral data files (Thermo.RAW) were searched with Proteome Discoverer 1.4 (Thermo Scientific) using SEQUEST search engine of the selected human component of UniProtKB/SwissProt database (http://www.uniprot.org/, version 2015–09). The following parameters were applied: Trypsin was selected as the enzyme and a maximum of two missed cleavages were permitted, precursor mass tolerance at ±15 ppm and fragment mass tolerance at 0.05 Da. Carbamidomethylation of cysteine was defined as a static modification. Oxidation of methionine and biotinylation of lysine and N-termini were set as variable modifications. All reported data were based on high-confidence peptides assigned in Proteome Discoverer with FDR < 1%.

The mass spectrometric data was searched with Proteome Discover 1.4 (Thermo Scientific) using the SEQUEST search engine against the UniProtKB/SwissProt human proteome (http://www.uniprot.org/, version 2015–09). Search parameters were set either as in PMID: 29568061(QE runs; BioID) (*Liu et al.,*

*2018*) or in PMID: 28054750 (Orbitrap Elite runs; AP-MS)(*Heikkinen et al., 2017*) All data was filtered to medium- (AP-MS) or high-confidence (BioID) peptides according to Proteome Discoverer FDR 5% or 1%, respectively. The lists of identified proteins were conventionally filtered to remove proteins that were recognized with less than two peptides and two PSMs. The high-confidence interactors were identified using SAINT and CRAPome as in *Liu et al., 2018*. Each sampl22e's abundance was normalized to its bait abundance. These bait-normalized values were used for data comparison and visualization.

## GUIDE-Seq

HEK293T cells were cultured in DMEM (Life Technologies) supplemented with 10% FBS, and penicillin/streptomycin at 37°C with 5% $CO_2$. For each replicate, 300,000 cells were transfected in 20 µl Solution P3 (Lonza) on a Lonza Nucleofector 4-D using the program CM-137, according to the manufacturer's instructions; 400 ng of pcDNA-pDEST-Cas9wt-POLD3, 150 ng of gRNA encoding plasmid (pU6-gRNA-HEK site 4), and 5 pmol of dsODN were transfected.

The blunt-ended dsODN used in our GUIDE-Seq experiments was the same as which was used in the original publication ( *Tsai et al., 2015*), and then was prepared by annealing the two modified oligonucleotides of the following compositions:

> 5'- P-G*T*TTAATTGAGTTGTCATATGTTAATAACGGT*A*T -3' and
> 5'- P-A*T*ACCGTTATTAACATATGACAACTCAATTAA*A*C -3'

P represents a 5' phosphorylation and * indicates a phosphorothioate linkage.

Genomic DNA was isolated using Qiagen DNeasy Blood & Tissue Kit (250) (#69506) and sheared with a Bioruptor Pico instrument (15 s ON, 90 s OFF, 7 cycles) to an average length of around 400–500 bp. The sheared gDNA were then end-repaired, A-tailed, and ligated to half-functional adapters, incorporating an 8-nt random molecular index according to the details of the GUIDE-Seq protocol provided in the original GUIDE-Seq paper (*Tsai et al., 2015*). Next, two rounds of nested anchored PCR, with primers complementary to the oligo tag, were used for target enrichment, as described in original publication (*Tsai et al., 2015*). Next, the samples were QC by BioAnalyzer 2100, DNA concentrations quantified by Qubit, and samples were pooled together and sequenced using the Illumina MiSeq instrument.

Data analysis was performed using the GUIDE-Seq analysis pipeline (*Zhu et al., 2017*) . We used custom scripts (https://bitbucket.org/valenlab/guide-seq-pold3; *Labun, 2021*; copy archived at swh:1:rev:ea73a563ebb6e6679709edeed4ba0cd077b4eece) to extract UMI from the demultiplexed fastq reads and trimmed adapters with cutadapt v2.8 (TTGAGTTGTCATATGTTAATAACGGTAT and ACATATGACAACTCAATTAAAC). Afterward, the data was aligned to the human genome (hg38) using bowtie2 v2.3.5.1 (with options –local –very-sensitive-local). The GUIDE-Seq v1.18 package from Bioconductor was used to call off-targets with the default settings. Final off-targets were normalized against control data (HEK293T cells transfected with dsODN only). Control data was processed in the same pipeline as the modified Cas9 samples.

## Amplicon sequencing

BJ-5ta cell lines expressing guides against RNF2 and ELANE loci were electroporated as described earlier. The DNA was extracted using the DNeasy Blood Tissue Kit (Qiagen, #69504) according to the kit instructions.

Library preparation was performed using a two-step PCR method. For the first PCR, a pair of target-specific primers was designed to amplify the 150 bp area surrounding the cutting site. Each target primer additionally includes an extension at the 5' end: for forward primers, this contains the Illumina Read1 primer sequence (see below, nucleotides in bold) and an 8 bp UMI (nucleotides underlined), and for the reverse primers, this contains the Illumina Read2 primer sequence only (see below, nucleotides in bold):

> RNF2 fwd 5'–3': **ACA CTC TTT CCC TAC ACG ACG CTC TTC CGA TCT** <u>NNN NNN NN</u>G ACA AAC GGA ACT CAA CCA T
> RNF2 rev 5'–3': **GTG ACT GGA GTT CAG ACG TGT GCT CTT CCG ATC T**TG TTC TAT TTA AGT TTT CAT GTT CT
> ELANE fwd 5'–3': **ACA CTC TTT CCC TAC ACG ACG CTC TTC CGA TCT** <u>NNN NNN NN</u>C TCC CCG GCA GAA ACG TC

ELANE rev 5'–3': **GTG ACT GGA GTT CAG ACG TGT GCT CTT CCG ATC T**GA GAA TCA CGA TGT CGT TGA GC

## Each PCR reaction was composed as follows

| Component | Final conc. | µl per one reaction |
|---|---|---|
| H₂O (milliQ) | | Add to obtain total of 50 µl |
| 5× Phusion GC Buffer (Thermo Fisher Scientific, #00928987) | 1× | 10 |
| 10 mM dNTPs (Thermo Fisher Scientific, #R0192) | 200 µM | 1 |
| 10 µM Primer fwd | 0.5 µM | 2.5 |
| 10 µM Primer rev | 0.5 µM | 2.5 |
| Betaine (Sigma Aldrich, #B0300) | 1 M | 10 |
| Phusion Hot Start II DNA Polymerase (2 U/µl) (Thermo Fisher Scientific, #00928987) | 0.02 U/µl | 0.5 |
| Template DNA | 500 ng | |

## Thermocycling conditions

| Cycle step | Temp. | Time | Cycles |
|---|---|---|---|
| 1. Initial denaturation | 98°C | 30 s | 1 |
| 2. Denaturation | 98°C | 10 s | 30 |
| 3. Annealing | 57°C | 10 s | |
| 4. Extension | 72°C | 20 s | |
| 5. Final extension | 72°C | 5 min | 1 |
| 6. Hold | 4°C | Forever | |

For the second PCR, the amplified products were purified using AMPure XP magnetic (Beckman Coulter, #A63882) according to the manufacturer's instructions, pooled and annealed with i5 and i7 Illumina Index primers. Both primers contain flow-cell-binding region (see below, highlighted in bold), index region (see below, underlined) and Illumina Read1 or Read2 primer binding regions, correspondingly (see below, italics): i5-PCR Index 11:

**AAT GAT ACG GCG ACC ACC GAG ATC TA**A CCT GGT T*AC ACT CTT TCC CTA CAC GAC GCT CTT CCG ATC\*T* i5-PCR Index 13:
**AAT GAT ACG GCG ACC ACC GAG ATC TA**C GGA ACA A*AC ACT CTT TCC CTA CAC GAC GCT CTT CCG ATC\*T* i7-PCR Index 13:
**CAA GCA GAA GAC GGC ATA CGA GAT** TTC CTC CT*G TGA CTG GAG TTC AGA CGT GTG CTC TTC CGA TC\*T* i7-PCR Index 14:
**CAA GCA GAA GAC GGC ATA CGA GAT** TGC TTG CT*G TGA CTG GAG TTC AGA CGT GTG CTC TTC CGA TC\*T* i7-PCR Index 15:
**CAA GCA GAA GAC GGC ATA CGA GAT** GGT GAT GA*G TGA CTG GAG TTC AGA CGT GTG CTC TTC CGA TC\*T* i7-PCR Index 16:
**CAA GCA GAA GAC GGC ATA CGA GAT** AAC CTA CG*G TGA CTG GAG TTC AGA CGT GTG CTC TTC CGA TC\*T*

## Each PCR was composed as follows

| Component | Final conc. | µl per one reaction |
|---|---|---|
| H₂O (milliQ) | | Add to obtain total of 50 µl |
| 5× Phusion GC Buffer (Thermo Fisher Scientific, #00928987) | 1× | 10 |

*Continued on next page*

*Continued*

| Component | Final conc. | µl per one reaction |
|---|---|---|
| 10 mM dNTPs (Thermo Fisher Scientific, #R0192) | 200 µM | 1 |
| 10 µM Primer i5 | 0.4 µM | 2 |
| 10 µM Primer i7 | 0.4 µM | 2 |
| Betaine (Sigma Aldrich, #B0300) | 1 M | 10 |
| Phusion Hot Start II DNA Polymerase (2 U/µl) (Thermo Fisher Scientific, #00928987) | 0.02 U/µl | 0.5 |
| Template DNA | 500 ng | |

## Thermocycling conditions

| Cycle step | Temp. | Time | Cycles |
|---|---|---|---|
| 1. Initial denaturation | 98°C | 30 s | 1 |
| 2. Denaturation | 98°C | 10 s | 10 |
| 3. Annealing | 58°C | 10 s | |
| 4. Extension | 72°C | 20 s | |
| 5. Final extension | 72°C | 5 min | 1 |
| 6. Hold | 4°C | Forever | |

PCR products were purified using AMPure XP magnetic beads (Beckman Coulter, #A63882) according to the manufacturer's instructions and DNA concentrations were measured with Qubit HS kit (Thermo Fisher Scientific, #2113695). Final products were pooled in equimolar concentrations into a single library with final concentration 250 nM (50× concentrate). Sequencing was done using an Illumina MiSeq v2 Micro flow cell, including 10% PhiX. The data analysis was performed using the AmpliCan software package (*Labun et al., 2019*).

## Statistics

Throughout the article text, we use the term 'replicates' to describe biological replicates (measurements of biologically distinct samples which represent biological variation), as opposed to the technical replicates (repeated measurements of the same sample to detect the variations stemming from measurement technique).

### High-throughput screening

This article describes one high-throughput screen:

(1) A screen with ~450 DNA repair proteins fused with Cas9 and tested in clonal HEK293T cells containing the GFP-BFP-sgRNA. The screen contained three biological replicates (=parallel cell culture wells in different culture plates) and was conducted twice (n = 2 × 3) (*Figure 1b*).

To calculate statistical significance, we first calculated the average of GFP+ cells in each biological replica. The data points were then compared to the combined mean of GFP+ cells in the whole screen ('experiment average') or the 48-well cell culture plate ('plate average'), as indicated in *Figure 1b*. We compared the mean of each replicate group to the combined mean of all other replicate groups using one-way ANOVA testing. The statistical parameters for these experiments are shown in *Supplementary file 4*. Note that in this case, unlike pairwise comparisons in usual ANOVA with multiple treatments, there is no multiple comparison problem since we compare in-turn the mean of each replicate group to a very large background group. Hence, each test statistic is dominated by the smaller replicate group so that the test samples are essentially disjoint.

For visualization, the data points were normalized to the average of GFP+ cells in the whole screen ('experiment average') or the 48-well cell culture plate average ('plate average'), as indicated in *Figure 1b*.

## Validation

*Figure 1c–d*: We chose 52 Cas9WT-DNA repair protein fusions that performed well in the main screen, and tested them in clonal HEK293T cells containing the GFP-BFP-sgRNA, with each tested fusion having four replicates (parallel cell culture wells in different 48-well plates). In addition, we tested the fusions in clonal RPE-GFP-RFP-sgRNA cells once (n = 1 × 2) and in clonal RPE-GFP-BFP-sgRNA cells once (n = 1 × 2). We used different clonal lines than that in the major screen to account for clone-specific phenomena.

*Figure 1f*: We chose 31 Cas9WT-DNA repair protein fusions that performed well in the main screen and in the first validation, and tested them in pooled HEK293T cells containing guides against the endogenous loci (ELANE, RNF2, Enh 4–1, CTCF1). Each tested fusion had three replicates (parallel cell culture wells in different 48-well plates) tested in four independent experiments (one independent experiment for each gene locus). For both validation experiments, the statistics were calculated similarly to the main screens.

*Figure 2b*: We chose 7 Cas9WT-DNA repair protein fusions that performed well in the previous validation experiments, and tested them in BJ-5ta-sgRNA cells containing guides against the endogenous loci (ELANE, RNF2, Enh 4–1, FANCF, STAT3). Each tested fusion had four replicates, except STAT3 where n = 3 (parallel cell culture wells in different six-well plates) tested in one independent experiment (one independent experiment for each gene locus).

*Figure 3b*: Four replicates (=parallel wells in a cell culture plate) for each condition (n = 1 × 4), one independent experiment. p-Values denote significance of the total editing (HDR + NHEJ) increment between the Cas9WT and other fusions (for 24–48 hr period). Statistical values derived using one-way ANOVA test.

*Figure 3e*: Five fluorescent microscopy images were taken from each individual glass slide (n = 5), each glass slide corresponds to one particular time point for each tested condition; data from one independent experiment. Statistical significance of the difference between Cas9WT and Cas9-POLD3 is calculated using ANOVA test for the equality of the means at a particular time point.

*Figure 3f–i*: One replicate ( = well in a cell culture plate) for each individual time point and electroporation cargo (n = 1), ddPCR individually conducted for each gene locus; data from one independent experiment (same set as shown in *Figure 3e*).

*Figure 4a–b*: The mass spectrometry experiments were conducted in biological duplicates (=parallel cell culture dishes). The data analysis is explained in the corresponding section.

*Figure 5a-b*: One replicate (=one well in a cell culture plate) for each condition (n = 1), one independent experiment.

*Figure 5c-f*: One replicate (=one well in a cell culture plate) for each condition (n = 1), one independent experiment.

*Figure 5g-h*: The most common NHEJ and HDR editing outcomes depicted in bar graphs (n = 3), one independent experiment. Bar denotes mean value, error bars represent ± SD. Statistical significance is calculated with unpaired, two-sided Student's t-test.

*Figure 6a*: Validation of Cas9-POLD3 against the panel of various HDR improving fusions in reporter HEK293T cells. n = 4, one of two independent experiments, bar denotes mean value, error bars represent ± SD. Statistical significance is calculated with unpaired, two-sided Student's t-test.

*Figure 6c*: Three Cas9 fusion proteins were tested in RPE-1 and HEK293T cells. n = 5, one of two independent experiments, bar denotes mean value, error bars represent ± SD. Statistical significance is calculated with ANOVA that is adjusted for multiple comparisons.

Summary of all statistical data is available in *Supplementary file 4*.

## Acknowledgements

Norwegian Research Council, the South-Eastern Norway Regional Health Authority, Knut and Alice Wallenberg Foundation, Cancerfonden, Barncancerfonden, Instrumentarium Foundation, and Academy of Finland supported this work. Part of this work was carried out at the High Throughput Genome Engineering Facility and the Swedish National Genomics Infrastructure funded by Science for Life Laboratory. We would like to thank the PSF at Karolinska Institutet for recombinant Cas9 production, and the HSØ Genomics Core Facility at Oslo University Hospital for providing the high-throughput sequencing. Sini Nieminen is acknowledged for expert technical assistance. We thank

Alicia Roig-Merino and Caoimhe Nic An tSaoir for their advice on MaxCyte electroporation experimental design.

## Additional information

### Funding

| Funder | Grant reference number | Author |
|---|---|---|
| Barncancerfonden | | Kornel Labun<br>Emma Haapaniemi |
| Norwegian Research Council | | Emma Haapaniemi |
| Ministry of Health and Care Services | 279922 | Emma Haapaniemi |
| Knut och Alice Wallenbergs Stiftelse | | Jussi Taipale |
| Cancerfonden | | Emma Haapaniemi |
| Instrumentariumin Tiedesäätiö | | Emma Haapaniemi |
| Science for Life Laboratory | | Bernhard Schmierer |
| Academy of Finland | | Markku Varjosalo<br>Jussi Taipale<br>Emma Haapaniemi |

The funders had no role in study design, data collection and interpretation, or the decision to submit the work for publication.

### Author contributions

Ganna Reint, Investigation, Validation, Visualization, Writing – original draft, Writing – review and editing; Zhuokun Li, Conceptualization, Methodology, Validation, Writing – review and editing; Kornel Labun, Eero Tolo, Formal analysis, Writing – review and editing; Salla Keskitalo, Leonardo A Meza-Zepeda, Susanne Lorenz, Investigation, Writing – review and editing; Inkeri Soppa, Katariina Mamia, Validation, Writing – review and editing; Monika Szymanska, Investigation, Project administration, Writing – review and editing; Artur Cieslar-Pobuda, Investigation, Resources, Writing – review and editing; Xian Hu, Diana L Bordin, Formal analysis, Resources, Writing – review and editing; Judith Staerk, Investigation, Methodology, Resources, Writing – review and editing; Eivind Valen, Data curation, Formal analysis, Writing – review and editing; Bernhard Schmierer, Markku Varjosalo, Investigation, Methodology, Writing – review and editing; Jussi Taipale, Conceptualization, Funding acquisition, Supervision, Writing – review and editing; Emma Haapaniemi, Conceptualization, Funding acquisition, Investigation, Methodology, Resources, Supervision, Writing – original draft, Writing – review and editing

### Author ORCIDs

Ganna Reint  http://orcid.org/0000-0003-4823-5485
Zhuokun Li  http://orcid.org/0000-0001-7297-6916
Monika Szymanska  http://orcid.org/0000-0003-0957-9568
Xian Hu  http://orcid.org/0000-0002-3381-7514
Judith Staerk  http://orcid.org/0000-0001-8698-6998
Bernhard Schmierer  http://orcid.org/0000-0002-9082-7022
Markku Varjosalo  http://orcid.org/0000-0002-1340-9732
Jussi Taipale  http://orcid.org/0000-0003-4204-0951
Emma Haapaniemi  http://orcid.org/0000-0002-6693-8208

### Decision letter and Author response

Decision letter https://doi.org/10.7554/eLife.75415.sa1
Author response https://doi.org/10.7554/eLife.75415.sa2

## Additional files

### Supplementary files
- Transparent reporting form
- Supplementary file 1. DNA repair domain library.
- Supplementary file 2. Mass spectrometry data.
- Supplementary file 3. Cas9 fusions - clustering of interactions.
- Supplementary file 4. Statistics.
- Supplementary file 5. CRISPR gRNA sequences.
- Supplementary file 6. FACS gating strategy.
- Supplementary file 7. ddPCR gating strategy and oligos.
- Supplementary file 8. Cas9 nuclease and GFP-BFP reporter cassette sequence.

### Data availability
The following data sets were generated: Sequence Read Archive (SRA), BioProject ID: PRJNA782085. The following previously published data sets were used: Tsai et al., (2015) Sequence Read Archive (SRA), SRP050338. Custom scripts used to extract UMI from the demultiplexed fastq reads for the GUIDE-Seq analysis is publicly available at: https://bitbucket.org/valenlab/guide-seq-pold3 (copy archived at https://archive.softwareheritage.org/swh:1:rev:ea73a563ebb6e6679709edeed4ba0cd077b4eece). Sequences of Cas9 nuclease and GFP-BFP reporter cassette used in this study are available in Supplementary file 8.

The following dataset was generated:

| Author(s) | Year | Dataset title | Dataset URL | Database and Identifier |
|---|---|---|---|---|
| Reint G, Li Z, Labun K, Keskitalo S, Soppa I, Mamia K, Tolo E, Szymanska M, Meza-Zepeda LA, Lorenz S, Cieslar-Pobuda A, Hu X, Bordin DL, Staerk J, Valen E, Schmierer B, Varjosalo M, Taipale J, Haapaniemi E | 2021 | Rapid genome editing by CRISPR-Cas9-POLD3 fusion | https://www.ncbi.nlm.nih.gov/bioproject/PRJNA782085 | NCBI BioProject, PRJNA782085 |

The following previously published datasets were used:

| Author(s) | Year | Dataset title | Dataset URL | Database and Identifier |
|---|---|---|---|---|
| Tsai SQ, Zheng Z, Nguyen NT, Liebers M, Topkar VV, Thapar V, Wyvekens N, Khayter C, Iafrate AJ, Le LP, Aryee MJ, Joung JK | 2015 | GUIDE-seq enables genome-wide profiling of off-target cleavage by CRISPR-Cas nucleases | https://www.ncbi.nlm.nih.gov/sra/SRP050338 | NCBI Sequence Read Archive, SRP050338 |

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
