## [Editor Report]

This is a well designed and well conducted study comparing ~450 human DNA repair protein and protein fragments in fusions with CRISPR/Cas9 for their efficiency in HDR genome editing and found that Cas9-POLD3 performed best in screening system or local targeting. The work proved that Cas9-POLD3 enhances editing at the early time points by speeding up the kinetics of Cas9 DNA binding and dissociation, allowing the rapid initiation of DNA repair. This work will be of interest to those wanting to improve gene editing efficiency by HDR.

---

## [Decision Letter]

[Editors' note: this paper was reviewed by Review Commons.]

---

## [Author Response]

Reviewer 1:This study performed a comprehensive comparison of 450 DNA repair protein – Cas9 fusions on HDR gene editing efficiency. A best fusion protein (Cas9-POLD3) was reported, and a possible mechanism and potential off-target effect of Cas9-POLD3 was studied using mass spectrometry and GUIDE-seq. Overall the key conclusions are solid and no obvious flaws are identified. The comments below can be largely addressed by the authors quickly.Major comments:The authors use ddPCR to quantify wild-type gene, correct gene and KO genes (Figure 1-2) as well as HDR and NHEJ (Figure 3). It's not clear how accurately ddPCR can quantify these events (wild-type, edited, HDR/NHEJ). It would be good to compare ddPCR quantifications with amplicon sequencing, which is considered a gold standard as it directly measures the editing outcomes.

ddPCR is a robust and reliable method for both HDR and NHEJ detection^1^. The 200-250bp area surrounding the cut is amplified using a set of specific primers and detection of the targeted amplicon is achieved via selective binding of the fluorescent reference probe. HDR detection is achieved via selective binding of a second fluorescent probe, matching only to the HDR-edited amplicons. NHEJ detection is achieved using a “drop-off” approach: a third fluorescent probe is designed to detect only the unedited DNA, and would not bind to the sequence with any acquired edits, resulting in the population of amplicons marked with the reference probe only (“total editing” population). Deduction of HDR values from the “total editing” values represents the NHEJ values (Figure S11).

To verify the reliability of the method, we have directly compared the editing readouts obtained using the ddPCR with the data from the amplicon sequencing (Figure S4K,L), using BJ-5ta cells edited with Cas9WT and Cas9-POLD3 mRNA. In this experiment, HDR and NHEJ values correlate reasonably well between the two methods. Even when ddPCR over- or underestimates true HDR or NHEJ efficiency for a given locus, the changes in editing efficiency remain reliable when the same gating is applied across all the samples during data analysis (data obtained for a related manuscript and thus not shown here). We have added references to manuscripts which successfully use ddPCR to quantify genome editing outcomes^1^,^2^. The text now reads:

“ddPCR is a sensitive and quantitative method to detect HDR and NHEJ at endogenous gene loci after gene editing^23^,^2^, and the results are comparable to Next Generation Sequencing-based genome editing quantification (Figure S4K,L).” (Page 4, lines 102-104).

Minor comments:It's recommended that the authors measure the expression changes upon Cas9-POLD3 introduction, as RNA-seq can tell how cells respond to Cas9-POLD3 fusion proteins.

We agree that RNA sequencing would provide useful information. As our sequencing core facility had a long queue for bulk RNA-seq, we chose instead to focus on the induction of p53 signaling^3^, as p53 alterations have the potential to increase the tumorigenic potential of the cells. For this experiment, we edited RPE-1 cells with a previously published RNF2 guide that has no detected off-targets^4^ using Cas9WT and Cas9-POLD3 plasmids. We collected the cells in four timepoints and analyzed the p21 (CDKN1A) expression by ddPCR (Figure S7). There is no significant difference in p53 activation between the treatment groups, suggesting that Cas9-POLD3 does not trigger an overt DNA damage response.

We have added this data in Figure S7 and in the main text, which now reads:

“CRISPR-Cas9 gene editing triggers p53-mediated DNA damage response^5^ that is proportional to the degree of DNA cutting^38^. […] We noted no difference in p53 activation between the treatment groups (Figure S7), suggesting that Cas9-POLD3 does not trigger an overt DNA damage response.” (Page 13, lines 242-246).

Significance (Required):The overall study is significant, as HDR repair has been one of the major approaches for precise gene editing. This study also systematically compares almost all of the protein – Cas9 fusions available, therefore providing very useful and important evaluations for potential users.The audience would be those who wish to use HDR for gene editing, including basic and clinical scientists. Therefore this study may reach a broad range of audience and the conclusions might be useful for further optimization of Cas9 fusion (for HDR) and for clinical applications (like gene therapy).I don't have sufficient to evaluate the interaction part (Figure 4) so the corresponding content was not evaluated.Referee Cross-commenting:No further comments. One thing that comes into my mind is the ddPCR issue I raised as a major point. I'm not entirely sure ddPCR can be used to directly evaluate NHEJ/HDR and wt/mutant allele, as I've never used ddPCR. If others think this is somewhat straightforward and not a concern, I can change that to a minor issue.Reviewer 2:This study by Reint et al. compared ~450 human DNA repair protein and protein fragments in fusions with CRISPR/Cas9 for their efficiency in HDR genome editing and found that Cas9-POLD3 performed best in screening system or local targeting. The work proved that Cas9-POLD3 enhances editing at the early time points by speeding up the kinetics of Cas9 DNA binding and dissociation, allowing the rapid initiation of DNA repair. In general, the study was well designed and conducted. I have raised the following concerns which hopefully can help clarify confusion and improve the manuscript.Major comments:1. The authors showed that the performance of Cas9-POLD3 was poorer that Cas9-Germini (Figure 6a). Why should we choose Cas9-POLD3 (I'm not convinced at this stage)? What is the advantage of using Cas9POLD3 if it is not significantly better than an existing one? Compared with Cas9-DN1S fusion which enhances HDR and inhibits NHEJ (Rajeswari Jayavaradhan et al., 2019), although the Cas9-POLD3 fusion enhances HDR, NHEJ is meanwhile enhanced simultaneously.

We added an experiment where we compared the previously published fusions CtIP, Geminin, HE+Geminin and HMGB1 in HEK293T and BJ-5ta cells in four endogenous loci (Figure 6B and Figure S8). DN1S and HE fusions were excluded due to the suboptimal HDR improvement in the initial GFP-reporter experiment (Figure 6A). We observe locus- and cell type specific variability and our findings are consistent with data reported in other publications^2,5^. The utility of the fusions thus varies based on the genomic context and experimental design. These observations underline the importance of the HDR-enhancing toolkit expansion.

The unoptimized application of the Cas9-POLD3 leads to the simultaneous increase of the NHEJ; however, it is possible to reduce this effect by fine-tuning the concentration of the Cas9-POLD3 reagent. Figure S5 illustrates that a 10-fold decrease in the concentration of Cas9-POLD3 plasmid does not affect the HDR values, but the NHEJ decreases approximately two-fold. Properly optimized setup gives on average 0.8 HDR/NHEJ coefficient for Cas9-POLD3, versus only 0.5 for Cas9WT. High performance of the Cas9POLD3 in lower concentrations also adds to the biosafety, since it will reduce the risk of the off-targets and regent-related cytotoxicity effects.

2. The authors have shown its improved performance in two human epithelium cell types (HEK293T and RPE1) and a human fibroblast BJ-5ta. The results in ESC and blood cells look somewhat disappointing. As a new method, it would also be interesting to demonstrate its wide applicability in other species, such as mouse. More interestingly, the authors may introduce protein engineering to screen a more powerful isoform of POLD3 if they are aiming for high impact journals.

We agree that a comprehensive testing of the fusions in a large panel of primary cell types would be useful. The fusions, however, need to be tested as mRNA (as in BJ-5ta cells), and in our hands, several primary cell types show very poor HDR editing in this setting (data not shown). These experiments would require the optimization of the Cas9 mRNA editing protocol, and we unfortunately lack the capacity to perform these large experiments in the context of this revision. Similarly, the comparison of several POLD3 isoforms and protein domains would indeed be interesting, but we lack the capacity to perform these experiments in the context of this revision. We have however noted this in the discussion by adding the following sentences:

“POLD3 fusion enhances editing by speeding up the initiation of DNA repair, and this effect might increase by using the individual domains of the POLD3 protein. In comparison to the previously reported fusions, POLD3 fusion provides a novel mechanism for editing enhancement, and combining POLD3 to protein domains with different editing-enhancing mechanisms might lead to additive benefit.” (Page 17, lines 327331).

We hypothesize, that poorer performance on the Cas9-POLD3 in the hESC and PBMCs could be partially explained by the faster cell division rates of these cell types (doubling rates: T cells – 10h^6^, hESC – 24h^7^, HEK293T cells – 24-30h^8^, BJ-5ta – 36h^9^). Faster division rate will account for more active physiological polymerase engagement along the DNA, which in turn will naturally hasten the removal of the Cas9 nucleases. Such predisposition could technically decrease the added benefit of the POLD3 fusion. Additionally, embryonic stem cells (ESC) lack a G1 checkpoint, have shortened G1/G2 cell cycle phases, and spend the majority of their time (up to 75%) in the S-phase of the cell cycle^10,11^. This physiological difference makes it easier for ESC to employ HDR in comparison to the differentiated cells, which spend at least half of their cell cycle in the G1 phase. Moreover, since genetic material damage in early embryonic development would have had the most devastating effects, DNA protection and reparation pathways are significantly enhanced in this cell type^12^. Such biological differences call for further investigations in terms of explorations of other Cas9 HDR-improving fusions.

3. The authors used GUIDE-seq to compare the off-target effect. However, I still suggest that the authors should perform RNA-seq to show that no significant changes of gene expression shall be observed when introducing POLD3.

We agree that RNA sequencing would provide useful information. As our sequencing core facility had a long queue for bulk RNA-seq, we chose instead to focus on the induction of p53 signaling^3^, as p53 alterations have the potential to increase the tumorigenic potential of the cells. For this experiment, we edited RPE-1 cells with a previously published RNF2 guide that has no detected off-targets^4^ using Cas9WT and Cas9-POLD3 plasmids. We collected the cells in four timepoints and analyzed the p21 (CDKN1A) expression by ddPCR (Figure S7). There is no significant difference in p53 activation between the treatment groups, suggesting that Cas9-POLD3 does not trigger an overt DNA damage response.

We have added this data in Figure S7 and in the main text, which now reads:

“CRISPR-Cas9 gene editing triggers p53-mediated DNA damage response^5^ that is proportional to the degree of DNA cutting^38^. […] We noted no difference in p53 activation between the treatment groups (Figure S7), suggesting that Cas9-POLD3 does not trigger an overt DNA damage response.” (Page 13, lines 242-246).

4. The AP-MS was conducted to find the interacting proteins of the Cas9-POLD3 complex to better understand the molecular mechanism of the improved performance of Cas9 HDR by POLD3. Can the authors respectively test the interacting proteins of Cas9? This would illustrate which key factors are brought in by POLD3 fusion.

We performed AP-MS for Cas9WT, and the interaction partners are included in the AP-MS plot. Cas9WT protein is indicated in pink, and the pink lines indicate Cas9WT interaction partners. Cas9-POLD3 fusion is indicated in green, and the green lines indicate Cas9-POLD3 interaction partners. Unique Cas9-POLD3 interaction partners are located within the bottom left cluster (marked “Unique interaction partners”), eg. MCM6 and MCM4 proteins. Rest of the clusters indicate shared interaction partners between Cas9WT and the studied fusions.

We have clarified the figure text, which now reads:

“Protein interaction map of the Cas9 fusions and Cas9WT. […] The Cas9 and gRNA were expressed from the genome.” (Pages 12-13, lines 223-229)

Significance (Required):This work systematically characterized the effect of ~450 human DNA repair protein and protein fragment fusions on CRISPR-Cas9 -based HDR genome editing, and discovered a subset of fusions that can enhance the genome HDR editing. The work provided an important reference data for researchers who want to improve HDR efficiency. The Cas9-POLD3 fusion performed best in the screening system, while tested as plasmids in reporter HEK293T line, HDR enhancement level is NOT better than the Geminin fusion. This diminishes the importance of the work. Compared with Cas9-DN1S fusion which enhances HDR and inhibits NHEJ (Rajeswari Jayavaradhan et al., 2019), although the Cas9-POLD3 fusion enhances HDR while NHEJ is enhanced simultaneously. So, Cas9-POLD3 is not a good candidate for HDR.

We added an experiment where we compared the previously published fusions CtIP, Geminin, HE+Geminin and HMGB1 in HEK293T and BJ-5ta cells in four endogenous loci (Figure 6B and Figure S8). DN1S and HE fusions were excluded due to the suboptimal HDR improvement in the initial GFP-reporter experiment (Figure 6A). We observe locus- and cell type specific variability and our findings are consistent with data reported in other publications^2,5^. The utility of the fusions thus varies based on the genomic context and experimental design. These observations underline the importance of the HDR-enhancing toolkit expansion.

The unoptimized application of the Cas9-POLD3 leads to the simultaneous increase of the NHEJ; however, it is possible to reduce this effect by fine-tuning the concentration of the Cas9-POLD3 reagent. Figure S5 illustrates that a 10-fold decrease in the concentration of Cas9-POLD3 plasmid does not affect the HDR values, but the NHEJ decreases approximately two-fold. Properly optimized setup gives on average 0.8 HDR/NHEJ coefficient for Cas9-POLD3, versus only 0.5 for Cas9WT. High performance of the Cas9POLD3 in lower concentrations also adds to the biosafety, since it will reduce the risk of the off-targets and regent-related cytotoxicity effects.

Reviewer 3:CRISPR/Cas-based precise genome editing utilizing the homology-directed repair (HDR) pathway can be inefficient in several cell types, mainly due to competition with the non-homologous end-joining pathway and due to cellular constrains in the HDR pathway and insufficient repair template availability. In this study from the Haapaniemi lab, the authors perform a systematic characterization of the HDR-enhancing effect of about 450 human DNA repair proteins fused to Cas9. The studies are mainly performed in HEK293T and RPE1 cells. For most protein fusions, there is no to little effects on the HDR rates, but the authors identify a Cas9 fusion to DNA polymerase δ subunit 3 (Cas9-POLD3) as a new fusion protein that enhances HDR rates. The authors show that this Cas9 fusion displays faster editing, does not increase offtarget rates or change the INDEL spectrum, and they show that it performs equally to or better than most xisting Cas9 fusion proteins (only exceeded by Cas9-Geminin fusion). In primary cells such as PBMCs and hESCs, the effect of Cas9-POLD3 mRNA delivery was less pronounced.The study is extensive, experiments are performed in adequate replicates, data is in general very well presented, and the writing is at a high standard. Even though the concept of fusing Cas9 to DNA repair modulators is not novel, this study is a comprehensive screen of around 450 different Cas9 fusions, which to my knowledge is unprecedented. The discovery that the Cas9-POLD3 fusion protein displays enhanced HDR rates over WT Cas9 is also novel. Combined, these two facts make this paper a highly significant contribution to the genome editing field.I have some points that could be addressed to further improve the article:I’m not sure I support the current title of the paper. Since the fusion protein screen is quite comprehensive, I believe the authors should include in the title that the results are from a screen. Furthermore, I suggest that it be stated that HDR is improved with Cas9-POLD3 and not only more rapid.

It is great that the reviewer acknowledges the comprehensiveness of the paper. We prefer to keep the original title because in our experience, succinct titles better catch the attention of the readers. We, however, are open to changing the title upon editor’s discretion to the following: “DNA repair protein screen identifies CRISPR-Cas9-POLD3 as a novel editing-enhancing fusion”.

It is stated that "HDR is confined to the synthesis (S) phase of the cell cycle (1)". I don't see that reference 1 actually has evidence of this. The general belief is that HDR is confined to S and G2 phases of the cell cycle. Please clarify and include proper references.

We have now corrected the reference to: “HR is restricted to the S and G2 phases of the cell cycle. Essers, J. et al. Analysis of mouse Rad54 expression and its implications for homologous recombination. DNA Repair 1, 779–793 (2002).”

We have also edited the sentence, which now reads:

“HDR is confined to the synthesis (S) and G2 phases of the cell cycle.” (Page 2, line 52).

I believe the statements in the introduction concerning the difficulty of obtaining HDR should be a little more balanced. There are examples in the literature of very high HDR efficiencies. For example, the authors could reference Wiebking et al. (PMID: 32241852) where a mean of 79.4% HDR is achived in primary human T cells from 11 different donors. Also, a recent publication from Lattanzi et al. (PMID: 34135108) shows 60-70% HDR in sickle cell patient-derived CD34+ HSPCs. This means, that under the right circumstances, HDR can be the dominant repair pathway and outcompete NHEJ and also lead to very high levels of HDR.

We have added these two references from the Porteus lab in the introduction, and the text now reads:

“In addition, recent studies show that providing the DNA repair template with long homology arms by a non-integrating rAAV6 can significantly increase HDR rates ^13^,^14^.” (Page 2, lines 57-59).

At the top of p.2 it would be useful to state some concrete examples from published literature of the editing efficiencies obtained with other Cas9 fusion strategies. What fold increases are observed in the literature and what are the absolute %HDR differences?

We have updated these in the introduction, which now reads:

“Enhancing CRISPR-based HDR by Cas9 fusion proteins can stimulate correction locally at the cut site, without causing generalized disturbance in the cellular DNA repair process and thus increasing the safety and specificity of the editing. […] The effect of the published fusions has been guide-, cell type- and locus-specific, reaching up to 30% (4-fold) HDR for Cas9-HN1HG1^21^, 20% (6fold) HDR for Cas9CtIP^19^,^20^ and 86% (3-fold) for Cas9-DN1S^22^, depending on the model system used.” (Pages 2-3, lines 65-70).

Throughout the text of the paper, information is generally lacking on delivery of Cas9, sgRNA, and repair template. Which delivery modalities were used? Cas9 plasmid, sgRNA plasmid, one single plasmid encoding both components or two separate plasmids, ssODN repair template or plasmid, lipofectamine, electroporation, etc. This info might be available in the Materials and methods sections and embedded into the figures, but it would be nice to have in the main text and the figures legends as well.

We have added this information in the text and figures legends.

P.4. Twice-fold should be two-fold I assume. Same sentence: how do you get 2-fold? And how is it independent of concentration? From Figure S5a is seems that for example at 30fmol the HDR increases from 5.5 to maybe 7.75. This would give a fold-increase of 1.4. Please clarify.

We have rewritten the text as “We further tested the performance of Cas9-POLD3 across decreasing concentrations in the RPE1 reporter cells and noted that in all four different concentrations, Cas9-POLD3 outperformed Cas9WT (Figure S5).” (Page 6, lines 134-136).

Figure 1E and 2A. Innocent SNPs? Do you mean synonymous (or silent) SNPs?

We meant silent SNPs and have updated the text in the figures.

Figure S5C. Why is ratio expressed as %. Wouldn't this usually be a regular number, for example 0.7?

We have corrected the mistake.

Figure 3C- Time course = time course.

We have corrected the mistake.

The last piece of data comparing the Cas9-POLD3 fusion to other published Cas9 fusion proteins is very informative. Also the data on RNP and mRNA delivery, which in general is less promising. However, I do miss some more extensive studies here. The data in Figure 6A is only from the GFP HEK reporter cell line.To get a more comprehensive characterization of the Cas9-POLD3 and how it compares to existing fusion variants, the authors should expand this data set to include more cell types and other endogenous loci. Furthermore, the RNP and mRNA delivery studies are also very small, performed only in HEK293 and RPE-1 reporter cells (RNP, GFP activation; one artificial locus) and in PBMCs and hESCs (mRNA, a single endogenous locus). I would also like to see RNP and mRNA performance in more cell types and at more endogenous loci. It would also be interesting to see how electroporation works for RNP activity instead of only cationic lipid transfer.

We have now compared the previously published fusions CtIP, Geminin, HE+Geminin and HMGB1 in HEK293T and BJ-5ta cells in four endogenous loci (Figure 6B and Figure 8). DN1S and HE fusions were excluded due to the suboptimal HDR improvement in the initial GFP-reporter experiment (Figure 6A). We observe locus- and cell type specific variability and our findings are consistent with data reported in other publications^2,5^. The utility of the fusions thus varies based on the genomic context and the experimental design.

We agree that a comprehensive testing of the fusions in a large panel of (primary) cell types would have been useful. In primary cells, the fusions need to be tested as mRNA as the plasmid expression is toxic. When we tried this, however, several primary cell types showed very poor HDR editing when transfected with Cas9 mRNA, gRNA and repair DNA (data not shown). We concluded that the experiments will require extensive optimization of the Cas9 mRNA editing protocol, which is outside the scope of this revision. We did not profile the fusions in cancer cell lines other than HEK293T, as cancer cell lines tend to edit well with standard Cas9WT RNPs. Cancer cell lines also have abnormal DNA repair signaling, which questions the utility of the results of the CRISPR fusion comparison for basic DNA repair biology.

Unfortunately, our protein production facility faced staffing challenges due to the pandemic, and we were not able to produce sufficiently pure Cas9-POLD3 protein during the scope of this revision.

Figure S9. I assume it’s crucial to state that one or both of the primers bind outside the homology arms. And to clarify the approach, it could be added that the NHEJ and HDR quantification occurs in two separate ddPCR reactions as far as I understand it.

Materials and methods section was modified to accommodate this specification. The text now reads:

“The fwd and reverse amplicon primers bind outside the homology arms of the repair DNA template. […] The HDR and NHEJ detection occurs in two separate ddPCR reactions.” (Page 25, lines 583-588).

The discussion states: "We have further identified a Cas9 fusion to POLD3, which improves editing by hastening the kinetics of DNA repair." I don't think there is solid evidence that this is the actual mechanism. This still seems like a hypothesis and the sentence should be modified accordingly.

We have rephrased the text as “We identify Cas9 fusion to POLD3 as a novel editing enhancer and propose the mechanism of action to be the rapid initiation of DNA repair.” (Page 17, lines 310-311).

In the discussion, it would be great if the authors could put their findings into a more therapeutic context. In which current or future therapies is there an unmet need to increase HDR? How can this fusion protein be applied to address this need?

We have added the following text:

“In comparison to the previously reported fusions, POLD3 fusion provides a novel mechanism for editing enhancement, and combining POLD3 to protein domains with different editing-enhancing mechanisms might lead to additive benefit. Rapid and efficient gene editing would be particularly useful when editing primary stem cells, which cannot be sustained for a long time in vitro, or in medical applications which require high precision and utilize diseased patient cells that are sensitive to generalized alterations in DNA damage signaling and chromatin remodeling.” (Page 17-18, lines 328-334).

In the Materials and methods, were the 450 expression constructs sequence-validated and if so how?

The Gateway entry clones were either picked from the orfeome or synthesized by GenScrip inc From the resulting expression clones, we sequence-validated one out of five 96-well clone plates. After screening, the top hits were validated by Sanger sequencing. We have added this in the Materials and methods, and the text now reads:

“Of the resulting five 96-well destination clone plates, we verified one plate by Sanger sequencing. After each screening round, the top hits were validated by Sanger sequencing.” (Page 23, lines 524-525).

Significance (Required):Included above.Referee Cross-commenting:No further comments. The other reviews seem to be in line with my own.

References:

1. Miyaoka, Y., Mayerl, S. J., Chan, A. H. and Conklin, B. R. Detection and Quantification of HDR and NHEJ Induced by Genome Editing at Endogenous Gene Loci Using Droplet Digital PCR. Methods Mol Biol 1768, 349-362, doi:10.1007/978-1-4939-7778-9_20 (2018).

2. Miyaoka, Y. et al. Systematic quantification of HDR and NHEJ reveals effects of locus, nuclease, and cell type on genome-editing. Scientific reports 6, 1-12 (2016).

3. Haapaniemi, E., Botla, S., Persson, J., Schmierer, B. and Taipale, J. CRISPR–Cas9 genome editing induces a p53-mediated DNA damage response. Nature medicine 24, 927-930 (2018).

4. Tsai, S. Q. et al. GUIDE-seq enables genome-wide profiling of off-target cleavage by CRISPR-Cas nucleases. Nature biotechnology 33, 187-197 (2015).

5. Charpentier, M. et al. CtIP fusion to Cas9 enhances transgene integration by homologydependent repair. Nature communications 9, 1-11 (2018).

6. De Boer, R. J., Homann, D. and Perelson, A. S. Different dynamics of CD4^+^ and CD8^+^ T cell responses during and after acute lymphocytic choriomeningitis virus infection. The Journal of Immunology 171, 3928-3935 (2003).

7. Panyutin, I. V., Holar, S. A., Neumann, R. D. and Panyutin, I. G. Effect of ionizing radiation on the proliferation of human embryonic stem cells. Scientific reports 7, 1-9 (2017).

8. https://web.expasy.org/cellosaurus/CVCL_0063.

9. https://web.expasy.org/cellosaurus/CVCL_6573. 10. avatier, P., Lapillonne, H., Jirmanova, L., Vitelli, L. and Samarut, J. Analysis of the cell cycle in mouse embryonic stem cells. Embryonic stem cells, 27-33 (2002).

11. Tichy, E. D. and Stambrook, P. J. DNA repair in murine embryonic stem cells and differentiated cells. Experimental cell research 314, 1929-1936 (2008).

12. Maynard, S. et al. Human embryonic stem cells have enhanced repair of multiple forms of DNA damage. Stem cells 26, 2266-2274 (2008).

13. Wiebking, V. et al. Genome editing of donor-derived T-cells to generate allogenic chimeric antigen receptor-modified T cells: Optimizing αβ T cell-depleted haploidentical hematopoietic stem cell transplantation. Haematologica 106, 847-858, doi:10.3324/haematol.2019.233882 (2021).

14. Lattanzi, A. et al. Development of β-globin gene correction in human hematopoietic stem cells as a potential durable treatment for sickle cell disease. Sci Transl Med 13, doi:10.1126/scitranslmed.abf2444 (2021).